# ECHO: TOWARDS ADVANCED AUDIO COMPREHENSION VIA AUDIO-INTERLEAVED REASONING

**Daiqing Wu**[1,3,7]  **Xuan Zhang**[3]  **Dongbao Yang**[1]  **Jiashu Yao**[3]  **Longfei Chen**[5]
**Qingsong Liu**[3]  **Sicheng Zhao**[6]  **Can Ma**[1]  **Yangyang Kang**[2,3,✉]  **Yu Zhou**[4,✉]

[1]IIE, Chinese Academy of Sciences  [2]Zhejiang University  [3]ByteDance China
[4]VCIP & TMCC & DISSec, College of Computer Science, Nankai University
[5]School of Information Science and Technology, ShanghaiTech University
[6]Department of Psychological and Cognitive Sciences, Tsinghua University
[7]University of Chinese Academy of Sciences

wudaiqing@iie.ac.cn  yangyangkang@bytedance.com  yzhou@nankai.edu.cn

## ABSTRACT

The maturation of Large Audio Language Models (LALMs) has raised growing expectations for them to comprehend complex audio much like humans. Current efforts primarily replicate text-based reasoning by contextualizing audio content through a one-time encoding, which introduces a critical information bottleneck. Drawing inspiration from human cognition, we propose audio-interleaved reasoning to break through this bottleneck. It treats audio as an active reasoning component, enabling sustained audio engagement and perception-grounded analysis. To instantiate it, we introduce a two-stage training framework, first teaching LALMs to localize salient audio segments through supervised fine-tuning, and then incentivizing proficient re-listening via reinforcement learning. In parallel, a structured data generation pipeline is developed to produce high-quality training data. Consequently, we present Echo, a LALM capable of dynamically re-listening to audio in demand during reasoning. On audio comprehension benchmarks, Echo achieves overall superiority in both challenging expert-level and general-purpose tasks. Comprehensive analysis further confirms the efficiency and generalizability of audio-interleaved reasoning, establishing it as a promising direction for advancing audio comprehension. Project page: https://github.com/wdqqdw/Echo.

## 1 INTRODUCTION

Audio serves as a primary medium for the expression, transmission, and reception of information in human communication (Hinton et al., 2012). As such, the capability to comprehend audio has become an essential objective in the development of embodied intelligence, as well as a key pursuit within the broader landscape of artificial general intelligence (Pfeifer & Bongard, 2006). Through integration of audio encoders with large language models (LLMs), large audio language models (LALMs) have evolved as a prominent paradigm for this goal. Notable systems, such as Salmonn (Tang et al., 2024) and Qwen2-Audio (Chu et al., 2024), exemplify this trend, demonstrating superior performance across fundamental audio tasks, including speech recognition (Panayotov et al., 2015), sound classification (Gong et al., 2022), and music analysis (Agostinelli et al., 2023).

Recently, reasoning capabilities in LLMs have witnessed rapid growth (Huang & Chang, 2022). By breaking down complex problems into intermediate steps, LLMs progressively derive solutions through a sequence of incremental results (Wei et al., 2022), leading to substantial improvements in previous challenging domains such as mathematics and coding (Liu et al., 2024; Hui et al., 2024). This development has spurred research on LALMs, motivating a shift from basic audio perception toward advanced audio comprehension, which entails nuanced interpretation and reasoning over real-world compound audio. Leveraging prompt engineering (Ma et al., 2025b), supervised fine-tuning (Xie et al., 2025), or reinforcement learning (Li et al., 2025a), a close wave of research succeeds in replicating text reasoning in LALMs. These approaches typically rely on a one-time encoding to contextualize audio content, after which reasoning unfolds exclusively in the text modality (Su et al., 2025), a format we refer to as **audio-conditioned text reasoning**. While this represents a significant milestone for LALMs, it inherently suffers from the gap between heterogeneous modalities. Unlike symbolic text, audio constitutes a continuous signal that carries more diverse and

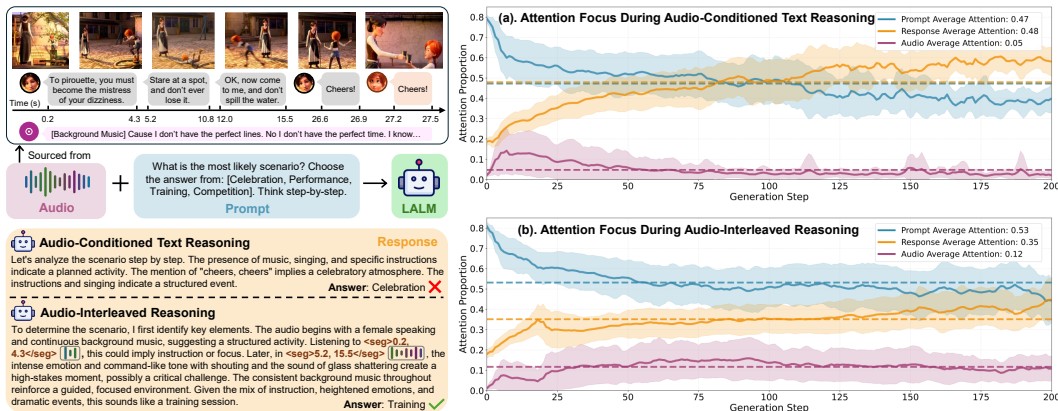

Figure 1: Comparison between audio-conditioned text reasoning and audio-interleaved reasoning. (a) and (b) compares attention allocated by the LALM to prompt, response, and audio tokens during reasoning, averaged over 100 samples. By switching from audio-conditioned text reasoning to audio-interleaved reasoning, LALM places significantly higher attention focus on the audio tokens (Δ+140%), thereby leading to more meaningful and traceable audio analysis.

fine-grained information. Consequently, one-time encoding imposes a severe information bottleneck—namely, the difficulty of preserving subtle audio details and reconstructing them from heavily compressed embeddings. This limitation risks LALMs overlooking salient details during reasoning, especially in realistic environments characterized by diversified and overlapping auditory sources.

In contrast, human auditory cognition involves cyclic re-listening of salient acoustic segments, driven by the interaction between auditory working memory and top-down attentional control (Griffiths & Warren, 2002; Oberauer & Hein, 2012). This mechanism enables humans to iteratively refine internal auditory representations and achieve more grounded and accurate decision-making (Bronkhorst, 2015; Nobre & Van Ede, 2018). Inspired by this mechanism, we introduce a new reasoning format termed **audio-interleaved reasoning**. It treats audio as active reasoning components rather than static contexts, granting LALMs direct access to acoustic content as needed. Consequently, LALMs overcome the previous information bottleneck and sustain deeper engagement with audio during reasoning. Figure 1 provides an intuitive comparison of the two reasoning formats. Under audio-conditioned text reasoning (a), LALM allocates a non-trivial share of attention (greater than 10%) to audio tokens only within the first 25 generation steps, after which this proportion rapidly falls below 5%. Conversely, audio-interleaved reasoning (b) stabilizes elevated attention to audio tokens (10%-14%) throughout the 25-200 steps. This sustained integration of auditory information enables the LALM to produce more logically coherent thoughts and more accurate responses to complex audio, as exemplified on the left.

To equip LALMs with audio-interleaved reasoning, we introduce a two-stage training framework. It begins with Qwen2.5-Omni (7B) (Xu et al., 2025), which is first enhanced through supervised fine-tuning to localize salient audio segments during reasoning. This yields a cold-start model capable of referencing specific temporal intervals via structured `<seg>` tag pairs and producing reasoning steps grounded in the corresponding audio segments. We denote this intermediate format as **audio-grounded reasoning**, which explicitly attends to the audio multiple times while remaining text-confined. In the second stage, we begin by directly interleaving raw audio tokens into the reasoning of the cold-start model. Each time a segment tag pair is decoded, the generation is paused. The corresponding audio tokens are then inserted immediately after the tag, forming an augmented context for subsequent generation. This extends reasoning into a genuinely multimodal process, which is further refined through reinforcement learning. Guided by verifiable reward signals, the model attains advanced proficiency in strategically re-listening to multiple audio segments, exploiting fine-grained information available in the raw audio to emulate human-like reasoning patterns.

Alongside this framework, we devise a structured data generation pipeline. It leverages audio datasets with available temporal metadata to produce tightly audio-grounded questions, answers, and chains of thoughts (CoTs). This process yields a curated collection of 75.9k audio question-answering (Audio-QA) samples with CoTs and 21.9k samples without, covering diverse audio categories and varying levels of difficulty. **Building on this resource, we present Echo, a LALM that instantiates audio-interleaved reasoning by proactively re-listening relevant audio segments**

**during the reasoning process.** When evaluated against other LALMs, Echo demonstrates superior overall performance on audio comprehension benchmarks (MMAR (Ma et al., 2025c), MMAU-mini, and MMAU (Sakshi et al., 2024)) that emphasize fine-grained interpretation and expert-level reasoning, even surpassing advanced proprietary systems such as GPT-4o (Hurst et al., 2024) and Gemini-2.0-Flash (Google, 2025). In summary, the contributions of this paper are threefold.

- Emulating human cognitive processes, we propose audio-interleaved reasoning, which regards audio as active components rather than static context. This formulation promotes sustained audio engagement, leading to more meaningful thoughts and precise answers.

- Supported by the two-stage training framework and data generation pipeline, we present Echo, a LALM that demonstrates exceptional expertise in dynamically localizing and re-listening to informative audio segments during reasoning.

- Systematic evaluation of Echo on three benchmarks comprehensively validates its effectiveness and efficiency, underscoring the viability and potential of audio-interleaved reasoning for advancing audio comprehension of LALMs.

## 2 RELATED WORK

### 2.1 LARGE AUDIO LANGUAGE MODEL

The success of LLMs constitutes a pivotal advance in artificial intelligence, catalyzing the emergence and growth of LALMs that leverage LLMs as the central orchestrator for general audio comprehension. Early systems such as AudioGPT (Huang et al., 2024) and HuggingGPT (Shen et al., 2023) mainly adopt a cascaded approach, which transcribes audio signals into text before feeding them into the LLM. To alleviate acoustic information loss, subsequent LALMs directly inject audio features into LLMs, typically by mapping them into the LLM embedding space via linear projection. This design has given rise to a wide spectrum of models, ranging from academic pioneers (Zhang et al., 2023; Deshmukh et al., 2023; Tang et al., 2024; Chu et al., 2023) to advanced commercial systems (Comanici et al., 2025; Goel et al., 2025; Wu et al., 2025a; Dinkel et al., 2025).

As expectations for LALMs evolve, research emphasis has increasingly turned toward handling the complexities of real-world compound audio. This trend is reflected in the recent emergence of more challenging audio benchmarks (Yang et al., 2024; Gao et al., 2024; Ma et al., 2025a; Ye et al., 2025). In particular, MMAU (Sakshi et al., 2024) and MMAR (Ma et al., 2025c) introduce intricate test cases that demand expertized interpretation and deliberate reasoning, aiming to quantify more sophisticated, human-like auditory intelligence. Building on these challenges, comprehensive evaluations across diverse tasks demonstrate that while existing LALMs achieve proficiency on basic perception tasks, they still face considerable limitations when confronted with expert-level audio questions, highlighting the pressing need for further advancement of audio comprehension.

### 2.2 MULTIMODAL CHAIN-OF-THOUGHT REASONING

The ability to perform CoT reasoning was first recognized (Kojima et al., 2022) and developed (Wang & Zhou, 2024) within LLMs and later extended to multimodal scenarios (Wang et al., 2025). In LALMs, Audio-CoT (Ma et al., 2025b) pioneers this direction through prompt engineering. Subsequent Audio-Reasoner (Xie et al., 2025) employs supervised fine-tuning to instill a structured reasoning pathway, adhering to the sequence of planning, caption, reasoning, and summary before deriving the final response. More recently, a growing body of work (Li et al., 2025a; Wen et al., 2025; Xing et al., 2025; Rouditchenko et al., 2025; Wu et al., 2025b) has shifted attention toward reinforcement learning, incentivizing reasoning abilities in LALMs with verifiable rewards.

Despite recent prosperity, current LALMs remain largely confined to audio-conditioned text reasoning. This format separates multimodal perception from the reasoning process, thereby imposing an intrinsic ceiling on the integration of nuanced multimodal cues. A similar bottleneck has recently prompted an evolution in the reasoning formats of large vision language models (LVLMs). Moving beyond "thinking about images" toward "thinking with images" (Su et al., 2025), modern LVLMs now seamlessly incorporate spatial references (Fan et al., 2025) or even raw image patches (Zheng et al., 2025; Zhang et al., 2025; Jiang et al., 2025) directly into their reasoning. This represents a

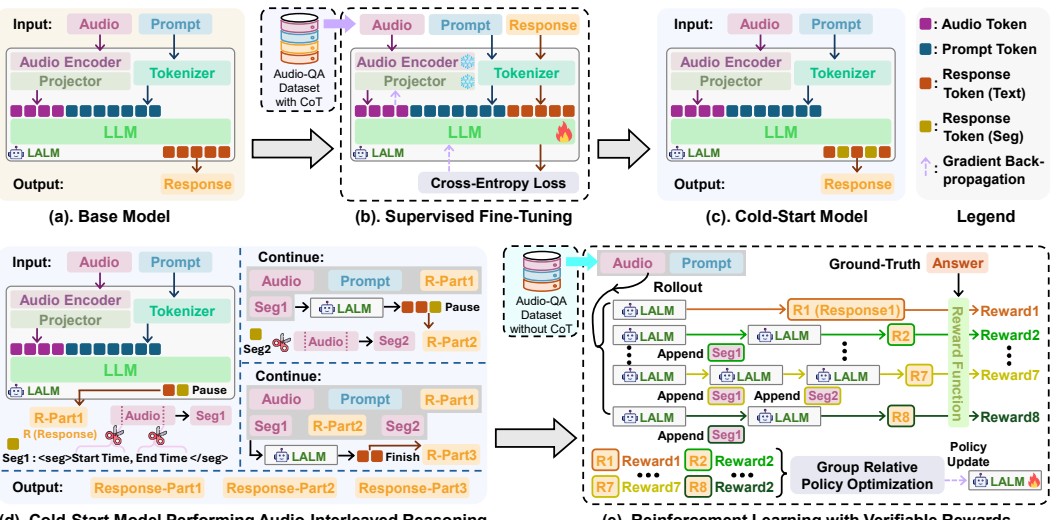

Figure 2: Summarized illustration of the training framework. The base model (a) is first enabled to localize and reference audio segments via SFT (b). The obtained cold-start model (c) is then equipped with audio-interleaved reasoning via inference adaptation (d): the inference process is paused whenever segment tags are encountered, and the corresponding raw audio segments are inserted afterwards before resuming. Subsequently, RL (e) is applied to further endow the model with competence in flexible audio invocation and accurate responding.

revolutionary step toward human-like reasoning (Larkin & Simon, 1987) and serves as a key inspiration for our work. Unlike images, audio is inherently temporal and can be naturally localized and segmented via timestamps. Capitalizing on this property, we propose audio-interleaved reasoning, an intuitive mechanism that empowers LALMs to dynamically localize salient audio segments and iteratively re-listen to them during reasoning. Through this formulation, we seek to catalyze an analogous evolution in LALMs: from "thinking about audio" to "thinking with audio".

## 3 ECHO

In this section, we provide a comprehensive introduction to **Echo**, detailing its two-stage training framework (Figure 2) and the accompanying data generation pipeline (Figure 3).

### 3.1 TRAINING FRAMEWORK

**The First Stage: Supervised Fine-tuning (SFT).** The training begins with a base model initialized from pre-trained Qwen2.5-Omni (7B) (Xu et al., 2025), denoted as $\pi_\theta$. When tasked with audio reasoning, the model demonstrates a persistent tendency to rely on natural language reasoning, being reluctant to incorporate specific audio segments to produce audio-grounded reasoning (Section C). Since this capability is crucial for localizing informative segments in audio-interleaved reasoning, we first endow the model with it through SFT.

Let the dataset be denoted as $\mathcal{D}_{\text{SFT}} = \{(x_i, y_i^*)\}_{i=1}^N$, which consists of $N$ high-quality Audio-QA pairs. Each sample $(x_i, y_i^*)$ includes a multimodal input $x_i = (\boldsymbol{A}, \boldsymbol{q})$, consisting of an audio $\boldsymbol{A}$ and a question $\boldsymbol{q}$, and a golden-standard response $y_i^* = [y_{i,1}^*, y_{i,2}^*, \cdots, y_{i,n}^*] = (\boldsymbol{c}, \boldsymbol{a})$, which is a $n$-length text composed of an audio-grounded CoT $\boldsymbol{c}$ enclosed in a `<think>` tag pair and a ground-truth answer $\boldsymbol{a}$ within a `<answer>` tag pair. The CoT is saturated with references to informative audio segments, with each structured as `<seg>`*start timestamp, end timestamp*`</seg>`. Each reference is deliberately preceded by a justification for its invocation and followed by a fine-grained and tightly associated analysis. Base model $\pi_\theta$ is subsequently optimized to proactively refer to informative audio segments and perform focused analysis during reasoning, by mimicking the demonstrations from $\mathcal{D}_{\text{SFT}}$ under the following cross-entropy objective:

$$\mathcal{L}_{\text{SFT}}(\theta) = -\frac{1}{n} \sum_{t=1}^{n} \log \pi_\theta(y_{i,t}^* | x_i, y_{i,<t}^*). \tag{1}$$

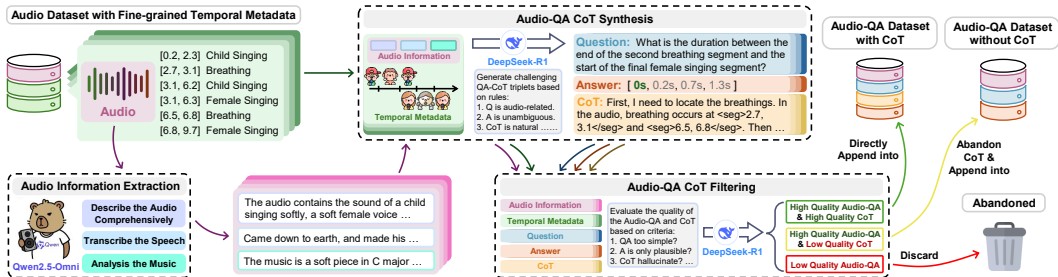

Figure 3: Overview of the data generation pipeline. It begins with an audio dataset containing fine-grained temporal metadata. For each audio data, Qwen2.5-Omni is employed to derive a structured caption encompassing comprehensive descriptions, speech content, and musical elements. This information, along with the temporal metadata, is then fed into DeepSeek-R1 (Guo et al., 2025) for synthesizing QA-CoT triplets. The synthesized triplets undergo further filtering based on the quality of Audio-QA and CoT, and are subsequently appended into two separate datasets for SFT and RL.

**The Second Stage: Reinforcement Learning (RL).** Upon acquisition of the cold-start model, still denoted as $\pi_\theta$, we first adapt its inference mechanism to activate audio-interleaved reasoning. Given an input $x_i = (\boldsymbol{A}, \boldsymbol{q})$, the model performs inference until a `<seg>` tag pair is detected. Let $s$ and $e$ represent the start and end timestamps encapsulated within the tag pair; the corresponding audio segment $\boldsymbol{A}_{s:e}$ is clipped from the initial audio $\boldsymbol{A}$. This segment, along with the current text output $\boldsymbol{o}$, forms an augmented input sequence $x_i' = (x_i \oplus \boldsymbol{o} \oplus \boldsymbol{A}_{s:e})$, which is fed back into the model to resume inference. This process repeats iteratively until a `<eos>` token is generated. The pseudocode is presented in Section D. Despite inheriting the capability to process multi-audio inputs from the base model, the cold-start model lacks sufficient competence in handling interleaved audio–text sequences. It restricts the model's effective attending to nearby audio segments during reasoning, obscuring the production of meaningful analysis coupled with fine-grained perception. To address this issue, we employ RL to facilitate more effective inference sampling (Yue et al., 2025), thereby unlocking the potential of audio-interleaved reasoning.

In reward design, several augmentations are applied to the standard format and accuracy rewards to accommodate the new reasoning format. Specifically, the base format reward $\mathcal{R}_{\text{format}}$ assigns 0.5 points to responses that correctly use the encapsulation tags. Based on empirical observations (Section J), we introduce an additional consistency reward $\mathcal{R}_{\text{consist}}$ to encourage semantic continuity after closing a `<seg>` tag pair. This reward checks whether the next token of `</seg>` is a capital letter (indicating the start of a new sentence) or the token < (suggesting continued segment reference or early termination of reasoning). A penalty of -0.1 is applied for either case, with the maximum cumulation to -0.5. On the other hand, the base accuracy reward $\mathcal{R}_{\text{acc}}$ allocates 0.5 points for answers that match the ground truth. Inspired by Zheng et al. (2025), we introduce a segment reward $\mathcal{R}_{\text{seg}}$ to incentivize audio segment re-listening. This reward grants an additional 0.5 points if the response refers to at least one segment and the answer is correct, and 0 points otherwise. In summary, the total reward for a response $\tau$ is computed as:

$$\mathcal{R}(\tau) = \mathcal{R}_{\text{format}}(\tau) + \mathcal{R}_{\text{consist}}(\tau) + \mathcal{R}_{\text{acc}}(\tau) + \mathcal{R}_{\text{seg}}(\tau). \quad (2)$$

Subsequently, Group Relative Policy Optimization (Shao et al., 2024) is employed for policy updation. Given an input $x$, the algorithm begins by sampling $G$ candidate responses $\{\tau_1, \tau_2, \cdots, \tau_G\}$ from the old policy model $\pi_{\text{old}}$. The advantage of response $\tau_g$ is then computed as the normalized reward across the mini-batch: $A_g = (\mathcal{R}(\tau_g) - \text{mean}(\mathcal{R}))/\text{std}(\mathcal{R})$. The cold-start model $\pi_\theta$ is updated to minimize a PPO-style (Schulman et al., 2017) clipped surrogate objective:

$$\mathcal{L}_{\text{RL}}(\theta) = -\frac{1}{G}\sum_{g=1}^{G}\frac{1}{|\tau_g|}\sum_{t=1}^{|\tau_g|}\left[\min(\rho_{g,t}(\theta)A_g, \text{clip}(\rho_{g,t}(\theta), 1\pm\epsilon)A_g) - \beta D_{\text{KL}}(\pi_\theta||\pi_{\text{ref}})\right]. \quad (3)$$

$\rho_{g,t}(\theta) = \pi_\theta(\tau_{g,t}|x, \tau_{g,<t})/\pi_{\text{old}}(\tau_{g,t}|x, \tau_{g,<t})$ serves as a correction factor to simulate on-policy sampled distribution. The clipping and the KL-divergence term constrain the model $\pi_\theta$ around the reference model $\pi_{\text{ref}}$. Notably, all interleaved audio tokens are ignored during loss computation.

Table 1: Accuracy comparison of advanced LALMs on MMAR. "Avg" denotes the macro-average accuracy across all audio types. Abbreviations: Sd = Sound, Ms = Music, Sp = Speech. **Bold**, underline, and wavy underline indicate the best, second-best, and third-best results, respectively.

| Model | Size | Single-Modality (%) | | | Mixed-Modality (%) | | | | Avg (%) |
|---|---|---|---|---|---|---|---|---|---|
| | | Sd | Ms | Sp | Sd-Ms | Sd-Sp | Ms-Sp | Sd-Ms-Sp | |
| Random Guess | - | 29.39 | 25.88 | 31.48 | 25.00 | 29.30 | 31.10 | 28.13 | 28.61 |
| *Open-Source Base LALMs* | | | | | | | | | |
| SALMONN (Tang et al., 2024) | 13B | 30.30 | 31.07 | 34.69 | 9.09 | 34.86 | 35.37 | 41.67 | 31.01 |
| Qwen-Audio (Chu et al., 2023) | 8.4B | 27.88 | 20.39 | 22.11 | 9.09 | 25.23 | 25.61 | 20.83 | 21.59 |
| GAMA (Ghosh et al., 2024) | 7B | 29.09 | 24.27 | 27.89 | 27.27 | 24.77 | 28.05 | 12.50 | 24.83 |
| Qwen2-Audio (Chu et al., 2024) | 8.4B | 33.33 | 24.27 | 32.31 | 9.09 | 31.19 | 30.49 | 25.00 | 26.53 |
| Baichuan-Omni-1.5 (Li et al., 2025b) | 11B | 41.21 | 33.01 | 40.48 | 36.36 | 48.62 | 39.02 | 41.67 | 40.05 |
| Audio-Flamingo-2 (Ghosh et al., 2025) | 3B | 24.85 | 17.48 | 20.75 | 18.18 | 26.61 | 23.17 | 8.33 | 19.91 |
| Qwen2.5-Omni (Xu et al., 2025) | 7B | 58.79 | 40.78 | 59.86 | 54.55 | 61.93 | 67.07 | 58.33 | 57.33 |
| *Proprietary LALMs* | | | | | | | | | |
| GPT-4o-mini-Audio (Hurst et al., 2024) | - | 38.79 | 35.92 | 58.84 | 45.45 | 60.09 | 57.32 | 50.00 | 49.49 |
| GPT-4o-Audio (Hurst et al., 2024) | - | 53.94 | 50.97 | 70.41 | 63.64 | **72.48** | 62.20 | **75.00** | 64.09 |
| Gemini-2.0-Flash (Google, 2025) | - | 61.21 | 50.97 | 72.11 | **81.82** | **72.48** | 65.85 | 70.83 | 67.90 |
| Gemini-2.5-Pro (Comanici et al., 2025) | - | 66.06 | 57.77 | **73.13** | **81.82** | 71.10 | 69.51 | 70.83 | **70.03** |
| *Adapted LALMs* | | | | | | | | | |
| Audio-CoT (Ma et al., 2025b) | 8.4B | 35.76 | 25.24 | 34.01 | 9.09 | 30.73 | 30.49 | 37.50 | 28.97 |
| Audio-Reasoner (Xie et al., 2025) | 8.4B | 43.64 | 33.50 | 32.99 | 45.45 | 42.66 | 31.71 | 25.00 | 36.42 |
| Omni-R1 (Rouditchenko et al., 2025) | 7B | 67.30 | 51.50 | 64.30 | 55.50 | 70.20 | 64.60 | 70.80 | 63.46 |
| Audio-Thinker (Wu et al., 2025b) | 7B | **68.48** | 53.88 | 64.29 | 72.73 | 71.56 | 73.17 | 66.67 | 67.25 |
| Echo (**Ours**) | 7B | 67.27 | **60.68** | 69.39 | **81.82** | 69.72 | **74.39** | 66.67 | 69.99 |

## 3.2 DATA GENERATION PIPELINE

Existing Audio-QA training datasets, such as AVQA (Yang et al., 2022) and Coltho-AQA (Lipping et al., 2022), are generally designed without a strong emphasis on task difficulty and lack golden-standard CoTs that incorporate fine-grained audio references. To address this limitation, we propose a data generation pipeline that leverages the advanced reasoning capabilities of LLMs to synthesize challenging Audio-QA data and corresponding audio-grounded CoTs for use in our RL and SFT. Figure 3 presents an overview of the pipeline.

We begin by selecting audio datasets enriched with available temporal metadata (in practice, AudioSet-Strong (Hershey et al., 2021) and MusicBench (Melechovsky et al., 2024)), and employ Qwen2.5-Omni to convert audio into textual descriptions. Each audio undergoes three independent procedures: once to generate a comprehensive caption, once to transcribe potential speech, and once to extract possible music elements. Combined with the temporal metadata, these outputs form a vivid textual simulation of the original audio. Based on this simulation, we prompt DeepSeek-R1 (Guo et al., 2025) to automatically synthesize QA–CoT triplets (refer to Section E for prompt usage). Benefiting from the advanced reasoning LLM, this process yields QA pairs that require fine-grained temporal analysis of the audio content, along with corresponding CoTs that follow a step-by-step reasoning process grounded in `<seg>` tag pairs to derive answers from the questions.

To further ensure the rationality and complexity of the Audio-QA pairs, as well as the faithfulness and logical coherence of the CoTs, the synthesized triplets are subjected to a rigorous re-evaluation by DeepSeek-R1 under detailed quality criteria. Triplets deemed to contain both high-quality Audio-QA and CoT are retained for the SFT dataset, while those with only high-quality Audio-QA are preserved without CoT and assigned to the RL dataset; all others are discarded. To further enrich the RL data, we perform direct quality evaluation for Audio-QA pairs from AVQA and select an additional 10k high-quality samples. In total, we obtain 75.9k high-quality Audio-QA samples with CoT, denoted as **EAQA-SFT** (Echo Audio-QA for SFT), and 21.9k Audio-QA samples without CoT, denoted as **EAQA-RL**. More comprehensive statistics are presented in Section F.

## 4 EXPERIMENTS

**Implementations.** For SFT, we use a learning rate of 5e-6 and a batch size of 16, freezing the audio encoder throughout training. The model is trained for one epoch on the EAQA-SFT dataset. For RL, we adopt a learning rate of 1e-6, a batch size of 64, a mini-batch size of 32, a KL-coefficient of 0.04, and 8 rollouts per query. The training proceeds for one epoch on the EAQA-RL dataset. During inference, Echo performs audio-interleaved reasoning following the identical adaptation detailed in

Table 2: Accuracy comparison of advanced LALMs on MMAU-mini and MMAU. "Avg" denotes the macro-average accuracy across three audio types.

| Model | Size | MMAU-mini (%) | | | | MMAU (%) | | | |
|---|---|---|---|---|---|---|---|---|---|
| | | Sd | Ms | Sp | Avg | Sd | Ms | Sp | Avg |
| Random Guess | - | 26.72 | 24.55 | 26.72 | 26.00 | 25.73 | 26.53 | 25.50 | 25.92 |
| Open-Source Base LALMs | | | | | | | | | |
| SALMONN (Tang et al., 2024) | 13B | 41.14 | 37.13 | 26.43 | 34.90 | 42.10 | 37.83 | 28.77 | 36.23 |
| GAMA (Ghosh et al., 2024) | 7B | 31.83 | 17.71 | 12.91 | 20.82 | 30.73 | 17.33 | 16.97 | 21.68 |
| Qwen2-Audio (Chu et al., 2024) | 8.4B | 67.27 | 56.29 | 55.26 | 59.61 | 61.17 | 55.67 | 55.37 | 57.40 |
| Qwen2.5-Omni (Xu et al., 2025) | 7B | 78.10 | 65.90 | 70.60 | 71.53 | 76.77 | 67.33 | 68.90 | 71.00 |
| Kimi-Audio (Ding et al., 2025) | 8.2B | 75.68 | 66.77 | 62.16 | 68.20 | 70.70 | 65.93 | 56.57 | 64.40 |
| DeSTA2.5-Audio (Lu et al., 2025) | 8B | 70.27 | 56.29 | 71.47 | 66.01 | 66.83 | 57.10 | 71.94 | 65.29 |
| Audio-Flamingo-3 (Goel et al., 2025) | 8.2B | 79.58 | 66.77 | 66.37 | 70.91 | 75.83 | **74.47** | 66.97 | 72.42 |
| Proprietary LALMs | | | | | | | | | |
| GPT-4o Audio (Hurst et al., 2024) | - | 64.56 | 56.29 | 66.67 | 62.51 | 63.20 | 49.93 | 69.33 | 60.82 |
| Gemini-2.0-Flash (Google, 2025) | - | 71.17 | 65.27 | 75.08 | 70.51 | 68.93 | 59.30 | 72.87 | 67.03 |
| Gemini-2.5-Pro (Comanici et al., 2025) | - | 75.08 | 68.26 | 71.47 | 71.60 | 70.63 | 64.77 | 72.67 | 69.36 |
| Step-Audio-2 (Wu et al., 2025a) | - | 84.04 | 73.56 | 75.15 | 77.58 | **80.60** | 68.23 | 72.75 | 73.86 |
| Adapted LALMs | | | | | | | | | |
| Audio-Reasoner (Xie et al., 2025) | 8.4B | 67.87 | 69.16 | 66.07 | 67.70 | 67.27 | 61.53 | 62.53 | 63.78 |
| Omni-R1 (Rouditchenko et al., 2025) | 7B | 81.70 | 73.40 | 76.00 | 77.03 | 78.30 | 70.80 | 75.80 | 74.97 |
| Audio-Thinker (Wu et al., 2025b) | 7B | 82.58 | 74.55 | 76.88 | 78.00 | 79.03 | 70.53 | 76.6 | 75.39 |
| Echo (**Ours**) | 7B | **86.49** | **76.35** | **78.38** | **80.41** | 79.62 | 72.33 | **77.87** | **76.61** |

Section 3.1 (Figure 2 (d)). We consider the answer correct only if the content within the `<answer>` tag pair matches exactly with the ground-truth, ignoring case and special characters.

**Evaluations.** Experiments are conducted on three popular Audio-QA benchmarks: MMAR (Ma et al., 2025c), MMAU (v05.15.25), and MMAU-mini (v05.15.25) (Sakshi et al., 2024), all of which are tailored for evaluating advanced audio comprehension. MMAR comprises 1k multi-disciplinary tasks requiring expert-level reasoning, spanning diverse audio types including speech, music, sound, and mixed-type content. MMAU and MMAU-mini place greater emphasis on general audio understanding, containing 9k and 1k tasks, respectively. They cover the evaluation across 27 distinct skills and three levels of task difficulty, each necessitating different degrees of reasoning over speech, music, and sound. Accuracy is employed as the primary evaluation metric across all benchmarks.

## 4.1 MAIN RESULTS

**Comparison on MMAR (Table 1).** When faced with reasoning-intensive audio tasks, most open-source base LALMs fail to demonstrate significant advantages over random guessing, revealing a pronounced gap between basic audio perception and advanced audio comprehension. In contrast, LALMs specifically adapted for the latter exhibit notable improvements. Collectively, Echo achieves the best average accuracy among open-source and adapted LALMs, even surpassing advanced proprietary LALMs such as GPT-4o-Audio and Gemini-2.0-Flash. These results underscore the benefit of decomposing complex audio inputs into simpler segments during reasoning, establishing audio-interleaved reasoning as a promising solution for challenging real-world scenarios.

**Comparison on MMAU-mini and MMAU (Table 2).** On general-purpose audio comprehension tasks, Echo also delivers competitive results across various types of audio inputs, achieving higher average accuracy (**+2.41%** on MMAU-mini and **+1.22%** on MMAU) than other open-source and proprietary LALMs. This outcome demonstrates that deeper engagement with audio content provides an inherent advantage for LALMs to produce perception-grounded analysis and reliable responses, confirming the generalizability of audio-interleaved reasoning in broader scenarios.

## 4.2 ANALYTICAL RESULTS

**Effectiveness of the Proposed Training Framework (Table 3: A→B→C→D).** From A→B, SFT results in a 4.97% accuracy gain, a 72% increase in average output length, and a corresponding 73% rise in reasoning latency. This effect stems primarily from the data distribution of EAQA-SFT, whose CoT annotations are produced by an advanced LLM and reflect a more elaborate, coherent, and audio-grounded reasoning path. Adapting the inference format to audio-interleaved reasoning (B→C) initially leads to a performance drop, as the model is not yet accustomed to interleaved audio–text inputs. However, subsequent RL (C→D) effectively mitigates this gap, with outcome-

Table 3: Comparison of average accuracy, response length, and reasoning latency on MMAR of models obtained from different training recipes. All inferences are obtained under a fixed prompt (Section E.2) and carried out on a single NVIDIA A100 GPU using the vLLM engine (Kwon et al., 2023). The proposed framework leading to Echo is represented by **A→B→C→D**.

| | Model | SFT Data | RL Data | Reasoning Format | MMAR (Avg) | | |
|---|---|---|---|---|---|---|---|
| | | | | | Acc (%) | Length (word) | Latency (s) |
| **A** | Base Model | - | - | Audio-Conditioned Text | 51.80 | 67.95 | 1.18 |
| **B** | Cold-Start Model | EAQA-SFT | - | Audio-Grounded | 56.77 | 117.06 | 2.04 |
| **B'** | Unadpted RL Model | EAQA-SFT | EAQA-RL | Audio-Grounded | 64.63 | 98.12 | 1.97 |
| **C** | Cold-Start Model | EAQA-SFT | - | Audio-Interleaved | 52.26 | 101.73 | 2.16 |
| **D** | Echo | EAQA-SFT | EAQA-RL | Audio-Interleaved | 69.99 | 107.40 | 2.12 |
| **D'** | Echo-AVQA | EAQA-SFT | AVQA | Audio-Interleaved | 67.58 | 120.18 | 2.37 |
| **E** | Direct RL Model | - | EAQA-RL | Audio-Conditioned Text | 63.15 | 104.42 | 2.06 |
| **E'** | Direct RL Model | - | AVQA | Audio-Conditioned Text | 59.51 | 111.14 | 2.11 |

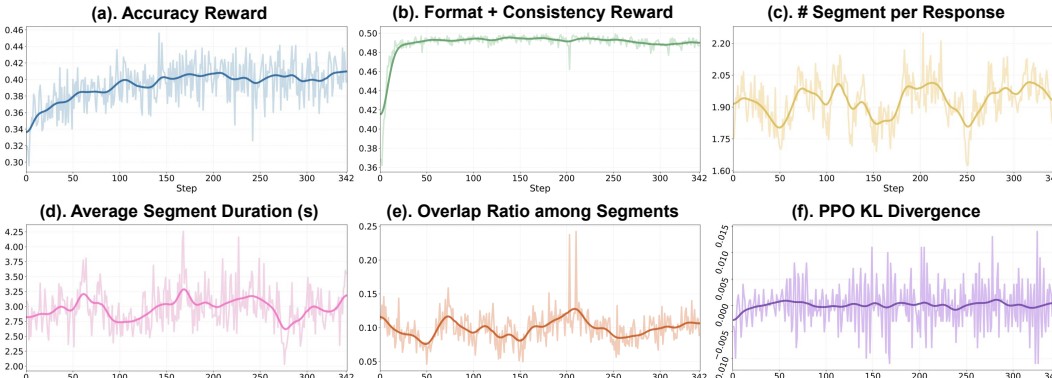

Figure 4: Evolvement of (a,b): reward components, (c,d,e): segment-associated statistics, and (f): model divergence during the RL process. (e) evaluates the temporal overlap among segments within each response. (f) measures the distribution discrepancy between the policy and the reference model.

based supervision ultimately boosting the average accuracy to a peak of 69.99%. Overall, these results validate the efficacy of our training framework, highlighting its role in unlocking the substantial potential of audio-interleaved reasoning.

**Comparison of Reasoning Format (Table 3: E→B'→D).** Lines **B'** and **E** represent models employing audio-grounded and audio-conditioned text reasoning formats, respectively, both subjected to the same RL process as Echo (**D**). Specifically, **B'** bypasses the inference adaptation from **B** to **C**, while **E** omits SFT, thus preserving the base model's original reasoning format. Along the trajectory **E→B'→D**, accuracy improves with increased audio involvement, underscoring the intrinsic benefit of audio-interleaved reasoning beyond data effects. Importantly, output length and inference latency remain comparable across formats, indicating that audio-interleaved reasoning incurs only a marginal computational trade-off, which guarantees its overall efficiency.

**Impact of RL Data (Table 3: D→D'& E→E').** Replacing the RL data from the 21.9k samples in EAQA-RL with 40.4k samples from AVQA results in a consistent decline in average accuracy across reasoning formats. It demonstrates that EAQA-RL delivers both superior annotation quality and more appropriately leveled challenges, thereby providing more effective supervision for incentivizing reasoning capabilities in LALMs.

**Training Dynamics of RL (Figure 4).** As training progresses, the accuracy reward (a) exhibits a fluctuating yet upward trend, and the format and consistency rewards (b) rapidly converge near their upper bounds. This demonstrates the effectiveness of verifiable rewards in both accuracy enhancement and format calibration. The efficacy of the segment reward is evident in segment-associated statistics: responses stabilize at invoking about 1.9 segments (c) with an average duration of 3.0s (d), and segment overlap (e) remains low around 0.1. These patterns indicate a healthy training process, in which the model consistently employs audio-interleaved reasoning with minimally overlapping segments. Finally, the PPO KL divergence (f) stays near zero, confirming that gradient clipping and the KL penalty effectively constrain the policy within a trusted region and ensure stability.

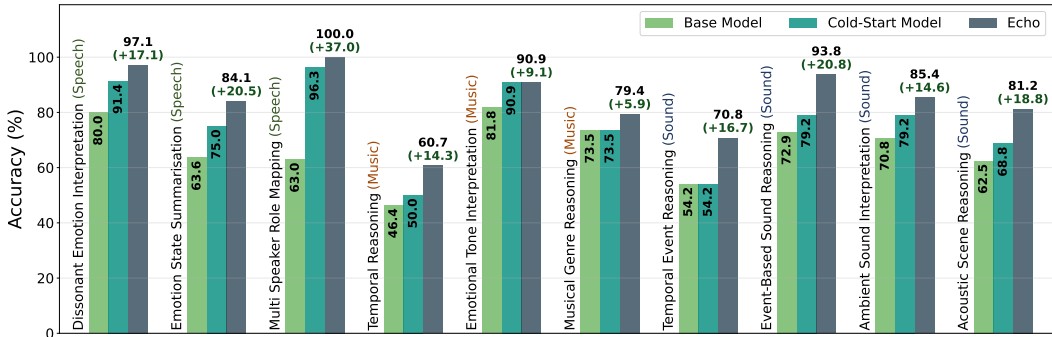

Figure 5: Progression of fine-grained cognitive abilities from the base model, to the cold-start model, and finally to Echo. The evaluation selects tasks from 10 representative skills, annotated in MMAU-mini as core requirements for LALMs, encompassing reasoning over speech, music, and sound.

**Skill-Wise Evolution on MMAU-mini (Figure 5).** A consistent enhancement is observed across 10 representative skills from the base model to Echo. The most substantial improvements occur in tasks associated with cognitive skills for speech and sound, exemplified by **+37.0%** on multi-speaker role mapping, **+20.8%** on event-based sound reasoning, and **+20.5%** on emotion state summarization. These tasks demand precise temporal localization and fine-grained audio perception, which are inherently strengthened by the format of audio-interleaved reasoning. Notable progress is also achieved in music-related skills. Though less dramatic, Echo attains consistent accuracy improvements exceeding 5%, demonstrating its competence in interpreting temporally dependent chord, melody, and genre. Collectively, these results confirm the broad effectiveness of audio-interleaved reasoning across diverse skill dimensions.

**Segment Coverage (Figure 6).** Echo demonstrates a strong proficiency in dynamically re-listening to audio segments. On 1000 tasks from MMAR, it re-listens to at least one audio segment in 99.4% of responses, two segments in 78.0%, and three or more in 28.4%. The re-accessed segments are distributed relatively evenly across the audio timeline, with a modest bias toward the first half. Interestingly, while EAQA-SFT annotations are constrained to the first 10 seconds due to limitations in source temporal metadata, 78.4% of MMAR audios extend beyond this range (further analyzed in Section K). Echo nevertheless identifies informative segments well past the 10-second mark, underscoring its ability to generalize segment localization across varying audio lengths.

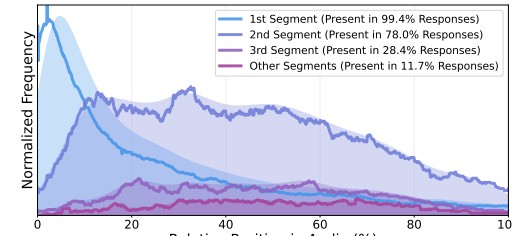

Figure 6: Distribution of segments coverage over relative audio positions in Echo's audio-interleaved reasoning to MMAR tasks.

## 5 CONCLUSION AND DISCUSSION

In this paper, we propose audio-interleaved reasoning, a new reasoning format that breaks through the information bottleneck inherent in conventional audio-conditioned text reasoning, offering a more human-aligned and flexible approach to audio comprehension. Building on it, we introduce a two-stage training framework consisting of SFT and RL, along with a data generation pipeline that produces accompanying EAQA-SFT and EAQA-RL datasets. These components collectively enable the development of Echo, a LALM equipped with audio-interleaved reasoning. Through extensive experiments, we demonstrate Echo's overall superior performance on audio comprehension tasks requiring expert-level reasoning. Further analyses reveal that this format promotes deeper audio engagement and enables accurate fine-grained perception, while incurring only minimal computational overhead. Overall, the results validate audio-interleaved reasoning as an effective path toward addressing challenging real-world scenarios and highlight its potential for future research.

Despite these advances, the current implementation in Echo remains relatively straightforward. There is considerable room for refinement through more customized audio re-listening or fine-grained supervision. For instance, while reaccessing original audio segments imitates human cognition, LALMs could potentially support more advanced manipulations, such as slow playback or

isolating specific frequency bands. Moreover, the CoT annotations in EAQA-SFT are automatically generated from fixed temporal metadata without human heuristics, and EAQA-RL similarly lacks detailed supervision over segment selection and usage. Explorations on these aspects could further unlock the potential of audio-interleaved reasoning.

# 6 ACKNOWLEDGEMENT

This work is supported by the National Natural Science Foundation of China (Grant No. 62406318 & 62376266 & 62571294 & 62441614) and the Beijing Natural Science Foundation (L252009).

# 7 ETHICS STATEMENT

This work advances the audio comprehension capabilities of LALMs by introducing a new reasoning format, a training framework, and corresponding datasets. The datasets are constructed from publicly available sources in compliance with their usage licenses, and no new data are collected. However, we note that the use of LLM (DeepSeek-R1) in the data generation process may inadvertently propagate biases inherent in its underlying training data. Specifically, as revealed by Marjanović et al. (2025), DeepSeek-R1 exhibits systematic "rumination", characterized by repeatedly revisiting similar problem framings rather than exploring diverse reasoning trajectories. The same study also highlights that DeepSeek-R1 may generate harmful or unsafe reasoning traces, which could, in turn, be unintentionally distilled into downstream models during training. Consequently, for downstream use in real-world applications, the dataset should undergo more rigorous processing to identify and mitigate such biases, thereby reducing potential negative impacts.

# 8 REPRODUCIBILITY STATEMENT

We provide a detailed description of our training framework (Section 3.1, Section D, Section E), data generation pipeline (Section 3.2, Section F), and evaluation settings (Section 4, Section G) to support the reproducibility of this research. We have released the training code and scripts in https://github.com/wdqqdw/Echo.

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

## A  APPENDIX MENU

## B  CASE ANALYSIS OF DIFFERENT REASONING FORMATS

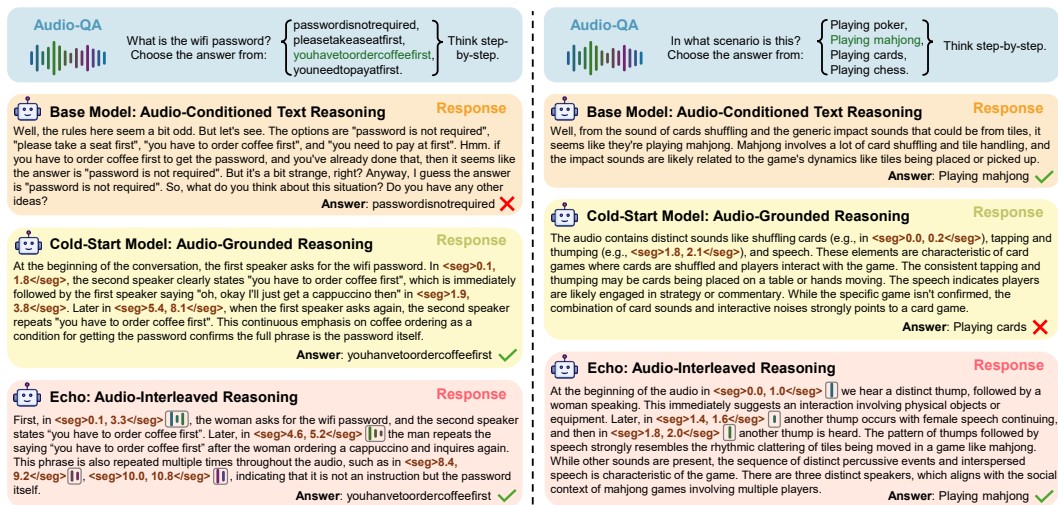

Figure 7: Intuitive comparison of reasoning formats on two representative MMAR tasks.

Figure 7 compares responses generated by different models, each employing a distinct reasoning format. The audio-conditioned text reasoning produced by the base model is entirely confined to natural language analysis, offering limited and superficial references to the audio content and failing to provide sufficient justification. After SFT, the cold-start model learns to generate audio-grounded reasoning that includes `<seg>` tag pairs along with traceable analyses. However, this mode remains essentially text-based and continues to overlook certain audio cues, occasionally leading the reasoning chain astray (as in the right case). In contrast, Echo's audio-interleaved reasoning dynamically inserts actual audio segments into the reasoning process. This facilitates deeper audio engagement and ultimately yields correct outcomes in both cases through logically coherent and perceptually grounded analysis.

## C  REASONING HABITS OF BASE MODEL

Figure 8 illustrates attempts to instruct the base model using different prompt templates to elicit audio-grounded reasoning. However, the base model consistently fails to adhere to these instructions and is reluctant to reference specific audio segments via timestamps. These results motivate our use of SFT to first endow the base model with this capability.

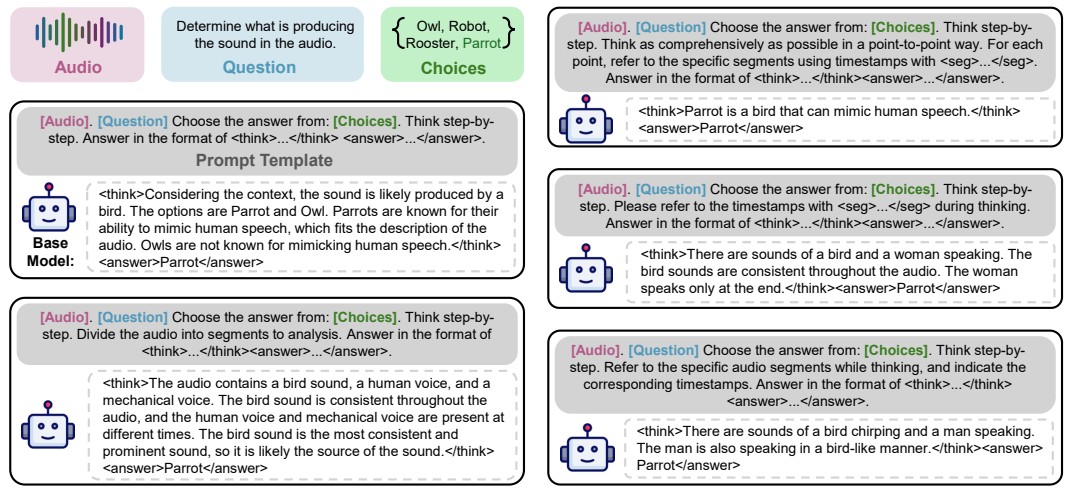

Figure 8: Responses of the base model under different prompt templates.

## D    PSEUDOCODE OF AUDIO-INTERLEAVED REASONING

Algorithm 1 presents the pseudocode of audio-interleaved reasoning.

---

**Algorithm 1** Inference Adaptation of Audio-Interleaved Reasoning

---

**Require:** Trained model $\pi_\theta$, input sequence $x_i = (\boldsymbol{A}, \boldsymbol{q})$
**Ensure:** Final output sequence $\boldsymbol{o}$
1: $\boldsymbol{o} \leftarrow \emptyset$
2: $\boldsymbol{A}_{\text{full}} \leftarrow \boldsymbol{A}$
3: $x \leftarrow x_i$
4: **repeat**
5:     $\hat{\mathbf{o}} \leftarrow \pi_\theta(x)$ {Perform inference until `<seg>` or `<eos>`}
6:     Append $\hat{\mathbf{o}}$ to $\boldsymbol{o}$
7:     **if** `<seg>` tag pair is detected in $\hat{\mathbf{o}}$ **then**
8:         Extract start timestamp $s$ and end timestamp $e$ from tags
9:         $\boldsymbol{A}_{s:e} \leftarrow \text{clip}(\boldsymbol{A}_{\text{full}}, s, e)$
10:         $x \leftarrow x \oplus \hat{\mathbf{o}} \oplus \boldsymbol{A}_{s:e}$ {Append output and audio segment}
11:     **else**
12:         $x \leftarrow \emptyset$ {Terminate if no segment tag}
13:     **end if**
14: **until** $\hat{\mathbf{o}}$ contains `<eos>` token or $x = \emptyset$
15: **return** $\boldsymbol{o}$

---

## E    PROMPT USAGE

### E.1    DATA CONSTRUCTION

**Audio Information Extraction: Comprehensive Caption**

Describe the audio as comprehensively as possible.

**Audio Information Extraction: Speech Transcription**

Give some information about the speech, such as the text content of the speech, the emotion of the speaker, the gender of the speaker, and the spoken language, only if the speech is present in this audio.

---

**Audio Information Extraction: Musical Elements**

Give some information about the music, such as music genre and music instruments, only if music is present in this audio.

---

**Audio-QA CoT Synthesis**

You are given simulated information about acoustic events in an audio clip, including:
A1: A description of the audio. A2: Information about speech (only if present), including possible transcript, emotion, and speaker gender. A3: Information about music (only if present), including genre and instruments. A4: A list of key audio segments where major sound events occur (with start and end timestamps). A5: The duration of the audio.

These are used to simulate your understanding as if you had listened to the actual audio. You MUST treat them as your internal interpretation of "the audio", and you MUST NOT reference A1–A5 or "the audio description" explicitly in your output.

Your goal is to generate challenging question–answer–chain-of-thought (CoT) triplets in the following format: [QUESTION_TEXT]: "A question that requires listening to the audio to answer accurately.", [MULTI_CHOICE]: [a, b, c, d], [ANSWER]: the correct answer, [COT]: [think]...your reasoning here...[/think][answer]...[/answer]

**REQUIREMENTS:**
**1. Question Design:** The question must be answerable only by listening to the audio. It should require non-trivial inference, going beyond surface-level perception. The question may either involve explicit temporal framing (e.g., asking what happened first, last, or at the same time) or be temporally neutral (e.g., asking about a specific sound, event, or emotion). However, the answer must require temporal reasoning—such as identifying the order of events, comparing the lengths of different segments, or detecting overlaps in time.
**2. Multiple Choice Options:** The correct answer must be unambiguously supported by the audio. The incorrect answers must be plausible, but clearly incorrect when grounded in the audio.
**3. Chain-of-Thought (CoT):** The reasoning process must be natural, fluent, and well-structured. Avoid using bullet points or numbering. Each time a specific piece of evidence from the audio is discussed, the relevant audio segment must be explicitly referenced first, using the format `<seg>start_second, end_second</seg>` (e.g., `<seg>3.3, 5.7</seg>`). For example: "In `<seg>0.4, 2.5</seg>`, the speaker raises their voice, suggesting urgency."
"By listening to `<seg>0.3, 1.9</seg>`, we can infer that a dog is barking in the background." Each CoT must include at least one segment reference, unless the provided segments all span the entire audio duration. General reasoning steps not tied to any specific moment do not require segment tags.
**4. Use of Simulated Inputs (A1–A5):** Treat A1–A5 as your internal hearing/understanding of the audio. Refer to any sounds or events by saying they occurred "in the audio". You MUST NOT refer to A1–A5 or "the audio description" as external annotations.

**Example:**

```
A1: The audio contains a speech across a range of times, a siren
    sound, and breathing.
A2: There is speech in Mandarin with a male speaker, expressing a
    fearful mood. The speech content is 'more than a copper are
    falling down here.
A3: There is no music in this audio.
A4: {"strong_events": [{"time_range": "0.0s – 1.4s", "label": "Male
    speech, man speaking"}, {"time_range": "0.0s – 2.0s", "label":
    "Wind"}, {"time_range": "1.7s – 2.0s", "label": "Tick"}, {"
    time_range": "2.1s – 10.0s", "label": "Wind"}, {"time_range":
    "2.4s – 2.6s", "label": "Tick"}, {"time_range": "3.0s – 3.1s", "
    label": "Tap"}, {"time_range": "3.2s – 3.4s", "label": "Tick"},
    {"time_range": "3.6s – 4.1s", "label": "Surface contact"}, {"
    time_range": "4.1s – 4.9s", "label": "Male speech, man speaking
```

```
    "}, {"time_range": "4.9s – 5.7s", "label": "Breathing"}, {"
    time_range": "5.9s – 6.0s", "label": "Tick"}, {"time_range":
    "6.0s – 6.7s", "label": "Male speech, man speaking"}, {"
    time_range": "6.3s – 10.0s", "label": "Emergency vehicle"}, {"
    time_range": "6.9s – 7.4s", "label": "Male speech, man speaking
    "}, {"time_range": "9.2s – 9.9s", "label": "Male speech, man
    speaking"}]]}
A5: 10.0
Output: {
  [QUESTION_TEXT]: How long after the surface contact sound ends
     does the man next speak?,
  [MULTI_CHOICE]: [0 seconds (immediately after), 0.3 seconds, 0.8
     seconds, 1.2 seconds],
  [ANSWER]: 0 seconds (immediately after),
  [COT]: [think]In <seg>3.6, 4.1</seg>, the surface contact occurs,
      ending at 4.1 seconds. Immediately after, in <seg>4.1, 4.9</
     seg>, the man begins speaking again. Since the speech starts
     at the exact moment the surface contact ends, there is no
     temporal gap between the two events. Therefore, the man speaks
      immediately after the surface contact ends.[/think][answer]0
     seconds (immediately after)[/answer],
}
```

**Input Format:**

```
A1: [DESCRIPTION]
A2: [SPEECH]
A3: [MUSIC]
A4: [SEGMENTS]
A5: [DURATION]
```

Do not output contents other than the question–answer–chain-of-thought (CoT) triplets.

---

## Audio-QA CoT Filtering

You are given simulated information about acoustic events in an audio clip, including:
A1: A description of the audio. A2: Information about speech (only if present), including possible transcript, emotion, and speaker gender. A3: Information about music (only if present), including genre and instruments. A4: A list of key audio segments where major sound events occur (with start and end timestamps). A5: The duration of the audio.
Your task is to evaluate the quality of the following Question–Choices–Answer–Chain-of-Thought (CoT) quadruplet based on the provided acoustic attributes (A1–A5).

**Evaluation Criteria:**
**1. Question and Answer Quality**

- **Necessity of Audio Information**: Does answering this question require listening to or analyzing the provided audio information (A1–A5)?

- **Answer Plausibility**: Is the correct answer (A) the only plausible choice among the given options?

- **Ambiguity**: Are there other choices that could be correct based on the given information? If yes, the QA is invalid.

**2. CoT Quality**

- **Grounding**: Does the CoT mention any information that is not present in A1–A5? If yes, consider the reasoning flawed.

- **Coherence**: Is the CoT logically coherent, step-by-step, and does it naturally lead to the given answer?

- **Fluency**: Is the reasoning presented in clear and natural English, without redundancy or contradiction?

**Audio Info:**

```
A1: [DESCRIPTION]
A2: [SPEECH]
A3: [MUSIC]
A4: [SEGMENTS]
A5: [DURATION]
```

**Output Format:**

```
[QA valid]: Yes/No, [COT valid]: Yes/No
```

Do not output any reason.

### E.2 EVALUATION

**Standard prompt template for evaluation of LALMs**

[QUESTION] Choose the answer from [CHOICES]. Think step-by-step. Refer to the specific audio segments while thinking, and indicate the corresponding timestamps. Answer in the format of `<think>...</think><answer>...</answer>`.

## F EAQA-SFT&EAQA-RL

### F.1 STATISTICS

Table 4: Comprehensive statistics of EAQA-SFT, EAQA-RL, and AVQA.

| Dataset | # Sample | Data Source | Avg Audio Length | Require Temporal Analysis | Include CoT Annotation | Average CoT Length | # Choices |
|---------|----------|-------------|------------------|---------------------------|------------------------|--------------------|-----------|
| EAQA-SFT | 75,862 | AudioSet-Strong (79.8%) MusicBench (20.2%) | 9.85s | ✓ | ✓ | 87.5 words | 4 (99.5%) >5 (0.5%) |
| EAQA-RL | 21,900 | AudioSet-Strong (7.5%) AVQA (46.8%) MusicBench (45.7%) | 9.86s | ✓ | ✗ | - | 2 (10%) 3 (5%) 4 (85%) |
| AVQA | 40,425 | AVQA (100%) | 9.96s | ✗ | ✗ | - | 4 (100%) |

Table 4 presents detailed statistics of the training datasets, including our constructed EAQA-SFT, EAQA-RL, and the widely adopted AVQA dataset. Benefiting from the temporal metadata of the source data and the advanced reasoning capability of DeepSeek-R1, both EAQA-SFT and EAQA-RL emphasize the fine-grained analysis of temporal clues. Additionally, EAQA-SFT is annotated with high-quality CoT data, with an average length of 87.5 words. EAQA-RL is designed with a varied number of choices to cover tasks of multiple difficulty levels.

### F.2 MITIGATION OF NOISES IN THE SOURCE METADATA

The temporal metadata in AudioSet-Strong and MusicBench inevitably contains annotation noise, which can propagate into the synthesized datasets and affect model training. Our data annotation pipeline reduces such noise through two complementary mechanisms. First, during the synthesis of QA-CoT triplets, the LLM (DeepSeek-R1) is provided with audio information from multiple sources, including temporal metadata and predictions from Qwen2.5-Omni. These sources implicitly cross-validate one another, allowing the LLM to assess the relative reliability of different cues. As illustrated in Figure 9, while the temporal metadata labels the audio segment as containing music and three sound effects, Qwen2.5-Omni detects no music and questions the third sound effect. When integrating these signals, the LLM synthesizes QA-CoT triplets only from information consistently supported across sources, thereby reducing the influence of uncertain temporal metadata. This implicit cross-validation mechanism promotes a more selective use of temporal annotations, thereby mitigating the propagation of noise into synthesized data.

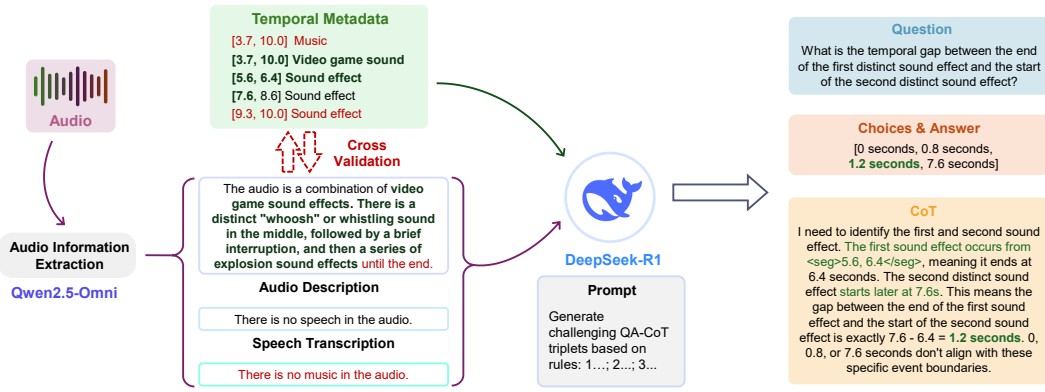

Figure 9: Audio information from multiple sources acts as implicit cross-validation for each other.

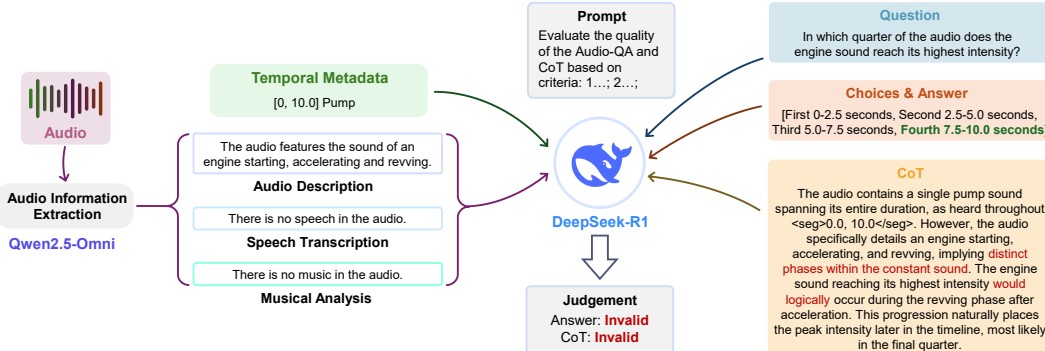

Figure 10: Re-evaluating process explicitly filters out identifiable hallucinated answers and CoTs.

Second, the LLM performs an additional round of re-evaluation to further improve the quality of the synthesized training data. In this stage, temporal metadata, audio information extracted from Qwen2.5-Omni, and the previously synthesized QA-CoT triplets are jointly provided to the LLM, allowing it to reassess each sample with full contextual evidence. Through explicit prompting, the LLM is instructed to verify answer plausibility, identify ambiguous choices, and detect unsupported or hallucinated CoT steps. As a result, implausible QA pairs and hallucinated CoTs are filtered out. As exemplified in Figure 10, an initial QA pair requires the model to determine sound intensity based on clues absent from both the temporal metadata and the audio information, leading to an implausible answer and ungrounded analysis in the CoT. By filtering similar cases, this stage further suppresses annotation noise and hallucinated reasoning in the final training sets.

Through these refinements, the noise present in the final EAQA-SFT and EAQA-RL datasets is substantially reduced. This improvement is empirically supported by the results in Table 3: the transitions (A→B) and (C→D) show clear performance gains on MMAR before and after training, and the comparison between **D** and **D'** further demonstrates the consistent advantage of EAQA-RL over the commonly used AVQA dataset.

### F.3  INFLUENCE OF SYNTHESIZER, RE-EVALUATOR, AND AUDIO EXTRACTOR

Table 5: Influence of substituting DeepSeek-R1 by other top-performed LLMs (Seed-1.6-Thinking (Seed, 2025) and GPT-5 (OpenAI, 2025)) for QA-CoT synthesizing and re-evaluating.

| Synthesizer | Re-evaluator | Datasets | # Sample | Average CoT Length | MMAR Acc (%) | MMAU-mini Acc (%) |
|---|---|---|---|---|---|---|
| DeepSeek-R1 | DeepSeek-R1 | EAQA-SFT EAQA-RL | 75,862 21,900 | 87.5 words - | 69.99 | 80.41 |
| Seed-1.6-Thinking | DeepSeek-R1 | EAQA-SFT-Seed EAQA-RL-Seed | 77,693 13,840 | 103.3 words - | 68.51 | 78.78 |
| DeepSeek-R1 | GPT-5 | EAQA-SFT-GPT EAQA-RL-GPT | 54,523 15,325 | 82.2 words - | 69.86 | 80.10 |

Table 5 compares the effects of using different LLMs to construct the training data. To ensure a fair evaluation, we maintain the identical training configuration across all SFT and RL experiments as introduced in Section 4 and Section G. Although substituting either the synthesizer or the re-evaluator leads to variations in sample numbers and CoT lengths, the downstream model performance remains largely consistent, which validates the stability and robustness of the proposed data generation pipeline. Notably, both variants yield a substantial reduction in the amount of RL data, yet this reduction leads to only marginal performance drops. This pattern suggests a potential trade-off between data quality and quantity: stricter filtering may increase the overall reliability of the retained samples, thereby enabling more data-efficient optimization.

Table 6: Influence of substituting Qwen2.5-Omni by other popular LALMs (Kimi-Audio (Ding et al., 2025) and Qwen2-Audio (Chu et al., 2024)) for aduio information extraction.

| Audio Extractor | Datasets | # Sample | Average CoT Length | MMAR Acc (%) | MMAU-mini Acc (%) |
|---|---|---|---|---|---|
| Qwen2.5-Omni | EAQA-SFT
EAQA-RL | 75,862
21,900 | 87.5 words
- | 69.99 | 80.41 |
| Kimi-Audio | EAQA-SFT-Kimi
EAQA-RL-Kimi | 83,387
19,569 | 94.5 words
- | 68.82 | 78.90 |
| Qwen2-Audio | EAQA-SFT-Qwen2
EAQA-RL-Qwen2 | 69,094
15,769 | 86.3 words
- | 63.91 | 75.41 |

Table 6 reports results from a similar setup, comparing the impact of different audio extractors. Unlike earlier observations, the choice of audio extractor leads to pronounced differences in data quality, as evidenced by downstream model performance. This underscores the pivotal role of the audio extractor in the data generation pipeline and suggests that it may currently constitute a bottleneck. From a broader perspective, improving the fidelity of audio information through stronger extractors effectively reduces labeling noise originating from temporal metadata. This result provides empirical support for the cross-validation mechanism described in Section F.2, offering additional insight into how the proposed pipeline mitigates noise in temporal annotations.

### F.4 INFLUENCE OF THE RE-EVALUATION PROCESS

Table 7: Ablation of the re-evaluation process.

| SFT Data | | RL Data | | MMAR | MMAU-mini |
|---|---|---|---|---|---|
| # Sample | Description | # Sample | Description | Acc (%) | Acc(%) |
| 75,862 | Re-evaluated as High Quality | 21,900 | Re-evaluated as High Quality | 69.99 | 80.41 |
| 111,868 | All Data Except for RL Data | 21,900 | Re-evaluated as High Quality | 64.67 | 76.19 |
| 75,862 | Re-evaluated as High Quality | 21,900 | Re-evaluated as Low Quality | 67.03 | 77.40 |

Table 7 provides a quantitative analysis of how the re-evaluation process influences downstream model performance. The first row represents the setting in which both the SFT and RL datasets consist exclusively of samples judged as high-quality during re-evaluation, and the model trained under this condition achieves the best overall performance. When low-quality samples, those that would ordinarily be filtered out, are introduced into the SFT data, downstream performance drops markedly. This result confirms the effectiveness of the re-evaluation stage in suppressing CoT noise and hallucinations. Substituting the RL data with samples identified as low-quality (we adopt this configuration mainly because scaling RL data substantially increases training cost) also leads to a decline in performance, though the degradation is less pronounced than in the SFT case. This may be attributed to the fact that the constructed QA samples generally contain fewer hallucinations and less noise compared to QA-CoT triplets.

## G  MORE IMPLEMENTATION DETAILS

Following the experimental setup described in Section 4, SFT is implemented using the ms-swift engine (Zhao et al., 2025), with a linear warm-up schedule applied during the first 5% of training steps. The RL process is conducted using the VERL engine (Sheng et al., 2025), with the sampling

temperature for rollouts set to 1.0. Model inference during rollout is performed using the vLLM engine (Kwon et al., 2023). For evaluation, a unified prompt template from Section E.2 is adopted across all experiments, along with a decoding temperature of 0.7. Line **A** in Table 3 represents our reproduced results, which are not entirely consistent with the results fetched directly from the original paper and reported in Table 2.

## H   DETAILS OF COMPARED BASELINES

In this section, we provide a detailed introduction to some current adapted LALMs to offer a comprehensive view of the field.

**Audio-CoT** (Ma et al., 2025b) represents a pioneering effort in exploring the reasoning capabilities of LALMs. It investigates three distinct CoT methods without modifying the model's parameters: 1). providing few-shot demonstrations; 2). adding the prompt "let's think step-by-step"; and 3). generating an audio caption before reasoning. Following Sakshi et al. (2024), we report results using the second approach in Table 1, with Qwen2-Audio (Chu et al., 2024) employed as the base LALM.

**Audio-Reasoner** (Xie et al., 2025) advances the reasoning capabilities of LALMs from a data-centric perspective. It utilizes Gemini-2.0-Flash and Gemini-2.0-Pro to construct CoTA, a large-scale dataset comprising 1.2 million Audio-QA samples with corresponding CoT. The CoT annotations in CoTA follow a structured pathway, including planning, captioning, reasoning, and summarization. Audio-Reasoner is obtained by performing SFT of Qwen2-Audio on this dataset. We did not adopt CoTA as the SFT dataset because its CoT annotations do not include <seg> tag pairs or accompanying segment-level analysis, which are essential for the development of the subsequent audio-interleaved reasoning.

**Omni-R1** (Rouditchenko et al., 2025) explores the impact of RL on LALMs. Although it does not specifically emphasize audio reasoning, it demonstrates that outcome-supervised GRPO is highly effective in optimizing LALMs. Notably, it shows that RL on pure Text-QA tasks can significantly enhance LALMs' performance on audio reasoning tasks. The study also introduces a pipeline for constructing QA pairs from raw audio data, which serves as an inspiration for our proposed data generation approach. In our comparison, we report its top-performing model, obtained by fine-tuning Qwen2.5-Omni on its generated VGGT-GPT dataset.

**Audio-Thinker** (Wu et al., 2025b), a very recent work, further advances the application of RL in LALMs. It guides the model to modulate its reasoning patterns during the GRPO process in response to tasks of varying difficulty levels, while employing an LLM-as-a-judge mechanism to provide explicit supervision signals for reasoning quality. Building upon Qwen2.5-Omni, it achieves state-of-the-art performance across MMAU, MMAR, and AIR benchmarks.

## I   QUANTITATIVE ANALYSIS OF RE-LISTENED SEGMENTS

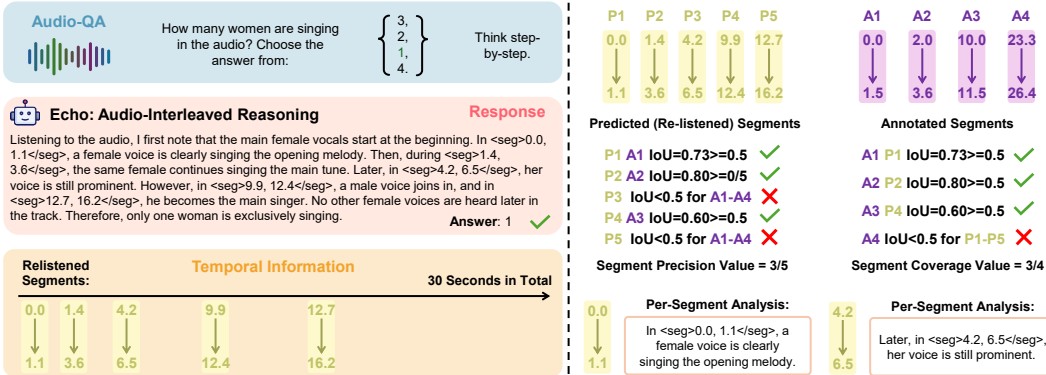

Figure 11: Illustration of how segment precision and segment coverage are computed for a sample, and how per-segment analysis is extracted for a segment.

## I.1 SEGMENT PRECISION AND SEGMENT COVERAGE

To quantitatively evaluate the accuracy of Echo in locating salient segments, we first introduce two sample-level evaluation metrics: segment precision and segment coverage. Both metrics rely on the Intersection over Union (IoU) between the predicted (re-listened) segments and the annotated segments. For two segments $A_{s_1:e_1}$ and $A_{s_2:e_2}$, where $s_1$, $s_2$ denote start times and $e_1$, $e_2$ denote end times, the IoU is calculated as:

$$\text{IoU}(A_{s_1:e_1}, A_{s_2:e_2}) = \frac{|(s_1, e_1) \bigcap (s_2, e_2)|}{|(s_1, e_1) \bigcup (s_2, e_2)|}. \tag{4}$$

Based on this, segment precision is defined as the proportion of predicted segments in a sample that have an IoU $\geq \rho$ with at least one annotated segment, reflecting the localization accuracy. Let the predicted segments for a sample be $\mathcal{P} = \{p_1, p_2, \ldots, p_n\}$, and the set of annotated segments be $\mathcal{A} = \{a_1, a_2, \ldots, a_m\}$. Segment precision at threshold $\rho$ is defined as:

$$P_s(\rho) = \frac{1}{|\mathcal{P}|} \Big| \{ p_i \in \mathcal{P} \mid \exists a_j \in \mathcal{A}, \text{IoU}(p_i, a_j) \geq \rho \} \Big|. \tag{5}$$

Conversely, segment coverage is defined as the proportion of annotated segments that have an IoU $\geq \rho$ with at least one predicted segment, indicating how well the predicted segments cover the annotated segments. Segment coverage at threshold $\rho$ is defined as:

$$C_s(\rho) = \frac{1}{|\mathcal{A}|} \Big| \{ a_i \in \mathcal{A} \mid \exists p_j \in \mathcal{P}, \text{IoU}(a_i, p_j) \geq \rho \} \Big|. \tag{6}$$

Figure 11 illustrates the computation process of these metrics with an example. Based on these metrics, we conduct a comprehensive evaluation of the model's proficiency in localizing audio segments using both our manually annotated and existing temporal metadata. In the absence of comparable baselines, we compute the average segment precision and segment coverage for both the cold-start model and Echo across three subsets: randomly selected samples, correctly answered samples, and incorrectly answered samples. This analysis offers insight into how RL influences segment-localization ability and how the quality of this localization, in turn, impacts the correctness of the final responses.

Table 8: Comparison of average segment precision and segment coverage on random, correctly answered, and incorrectly answered samples from MMAR.

| Model | Split | $\rho=0.3$ | | $\rho=0.5$ | |
|---|---|---|---|---|---|
| | | $P_s$ | $C_s$ | $P_s$ | $C_s$ |
| Cold-Start Model | Random | 88.32 | 70.62 | 81.21 | 59.68 |
| | Correctly Answered | 91.03 | 78.29 | 83.85 | 67.75 |
| | Incorrectly Answered | 86.79 | 63.85 | 78.19 | 52.32 |
| Echo | Random | 92.47 | 73.22 | 84.41 | 62.04 |
| | Correctly Answered | 94.25 | 79.26 | 85.38 | 68.98 |
| | Incorrectly Answered | 91.84 | 67.99 | 82.35 | 56.93 |

On MMAR, we select 50 correctly answered samples, 50 incorrectly answered samples, and 50 non-overlapping random samples, and manually annotate the audio segments that are critical for answering each question. These human annotations serve as the reference for evaluating average segment precision and segment coverage, with the results summarized in Table 8.

Echo achieves high performance on both metrics, indicating strong proficiency in segment localization. For correctly answered samples, Echo consistently exceeds its performance on incorrect samples in both precision and coverage, revealing a clear positive correlation between accurate segment localization and effective audio comprehension. The gap in segment coverage is particularly notable, suggesting that some model errors arise not from mislocalizing segments but from failing to retrieve essential regions of the audio.

The cold-start model exhibits localization ability that is already comparable to Echo's, though RL still provides measurable gains. This pattern suggests that outcome-based rewards implicitly improve segment-grounding behavior, even without explicit supervision on segment localization.

Table 9: Comparison of average segment precision and segment coverage on samples from Audio-QA constructed upon AudioGrounding (Xu et al., 2021).

| Model | Split | $\rho$=0.3 | | $\rho$=0.5 | |
|---|---|---|---|---|---|
| | | $P_s$ | $C_s$ | $P_s$ | $C_s$ |
| Cold-Start Model | All 500 Samples | 84.37 | 60.27 | 77.67 | 51.89 |
| | Correctly Answered | 90.53 | 66.70 | 81.98 | 55.01 |
| | Incorrectly Answered | 75.02 | 53.16 | 75.04 | 45.77 |
| Echo | All 500 Samples | 89.34 | 64.83 | 81.23 | 55.58 |
| | Correctly Answered | 91.65 | 69.71 | 83.88 | 59.75 |
| | Incorrectly Answered | 86.19 | 60.04 | 80.19 | 51.86 |

Since manual annotations inevitably introduce subjectivity, we further validate our conclusions using existing temporal metadata. The AudioGrounding dataset (Xu et al., 2021), originally designed for the text-to-audio grounding task, contains manually annotated sound event timestamps with relatively low noise. However, it does not provide QA pairs by default, so we apply our proposed data generation pipeline to construct 500 QA samples from its audio and event annotations.

In addition to overall evaluating all constructed samples, we also sample 50 correctly answered and 50 incorrectly answered cases, with results reported in Table 9. The overall trend is consistent with our earlier findings, though all metrics are slightly lower, especially segment coverage. This decline occurs because the segment labels in AudioGrounding are not filtered for task relevance, thus including segments unrelated to the QA task, which reduces measured coverage. Despite such differences, the consistent pattern across datasets validates our analysis.

## I.2 VALIDITY OF PER-SEGMENT ANALYSIS

To assess the plausibility of the model's analysis for each referenced segment, we adopt an LLM-as-a-judge evaluation. For each segment reference, we first extract the full sentence in which it appears (as exemplified in Figure 11). This sentence, together with the corresponding Audio-QA pair and the model's complete response, is then provided to an advanced LALM (Gemini-2.5-Pro in our implementation). The model is instructed to evaluate whether the analysis expressed in the sentence is relevant to the content of the referenced audio segment and whether it contributes meaningfully to the final answer. The prompt used for this assessment is shown below.

---

**Segment-level Analysis Plausibility Evaluation**

You are given the following inputs for evaluating the plausibility of a model's segment-level analysis:
S1: The full sentence in which a referenced audio segment appears. S2: The corresponding Audio-QA pair (question, choices, and correct answer). S3: The model's complete response for that Audio-QA instance.
Your task is to assess whether the analysis expressed in S1 is relevant to the referenced audio segment and whether it meaningfully contributes to deriving the correct answer.

**Evaluation Criteria:**

- **Relevance**: Whether the analysis in S1 is directly related to the content of the referenced audio segment.
- **Contribution**: Whether the analysis provides meaningful support for the final answer.

**Output Format:**

```
Relevance: Yes/No, Contribution: Yes/No
```

Do not provide explanations or reasoning.

---

We treat both relevance and contribution as segment-level metrics, assigning each a binary judgment (1 or 0). These metrics are computed on the segments drawn from the same 150 MMAR samples used in Table 8, and the aggregated results are reported in Table 10.

Table 10: Relevance and contribution of per-segment analysis on MMAR.

| Model | Split | Relevance | Contribution |
|-------|-------|-----------|--------------|
| Cold-Start Model | Random | 86.00 | 72.00 |
| | Correctly Answered | 86.00 | 78.00 |
| | Incorrectly Answered | 82.00 | 64.00 |
| Echo | Random | 94.00 | 84.00 |
| | Correctly Answered | 94.00 | 88.00 |
| | Incorrectly Answered | 92.00 | 72.00 |

The results indicate that, regardless of whether the final answer is correct or incorrect, Echo consistently produces highly relevant segment-level analyses, substantially outperforming the cold-start model. This highlights the advantage of audio-interleaved reasoning over an initial audio-grounded approach: re-listening to key segments enables Echo to deliver more accurate and context-aware interpretations. In contrast, the contribution metric exhibits a clear separation between correctly and incorrectly answered samples. Together with the findings from Section I.1, this pattern suggests a potential failure mode: Echo may sometimes anchor its reasoning on suboptimal segments. Even when the analysis of such segments is itself accurate, their weak alignment with the question can still mislead the final decision.

Overall, Echo attains high performance on both relevance and contribution, confirming that the model is capable of producing precise and reliable per-segment analyses as intended.

## J  ABLATION STUDY OF REWARD DESIGN

### J.1  IMPACT OF CURRENT REWARD COMPONENTS

Table 11: Comparison of models undergoing RL of different reward formulation. "Include" evaluates whether a response contains any segment reference. "Coverage" measures the non-overlapping total duration of all segment references in each response and their proportion relative to the audio duration.

| | Reward Formulation | MMAR (Avg) | | | | |
|---|---|---|---|---|---|---|
| | | Acc (%) | Length (word) | Segment | | |
| | | | | Include (%) | Count | Coverage (s)/(%) |
| A | Accuracy + Format | 67.84 | 77.27 | 86.40 | 1.62 | 3.94 / 23.32 |
| B | Accuracy + Format + Consistency | 68.07 | 113.37 | 88.50 | 2.11 | 4.59 / 27.15 |
| C | Accuracy + Format + Segment | 68.22 | 86.58 | 98.70 | 2.11 | 4.94 / 29.42 |
| D | Accuracy + Format + Consistency + Segment | 69.99 | 107.40 | 99.40 | 2.25 | 4.17 / 25.08 |

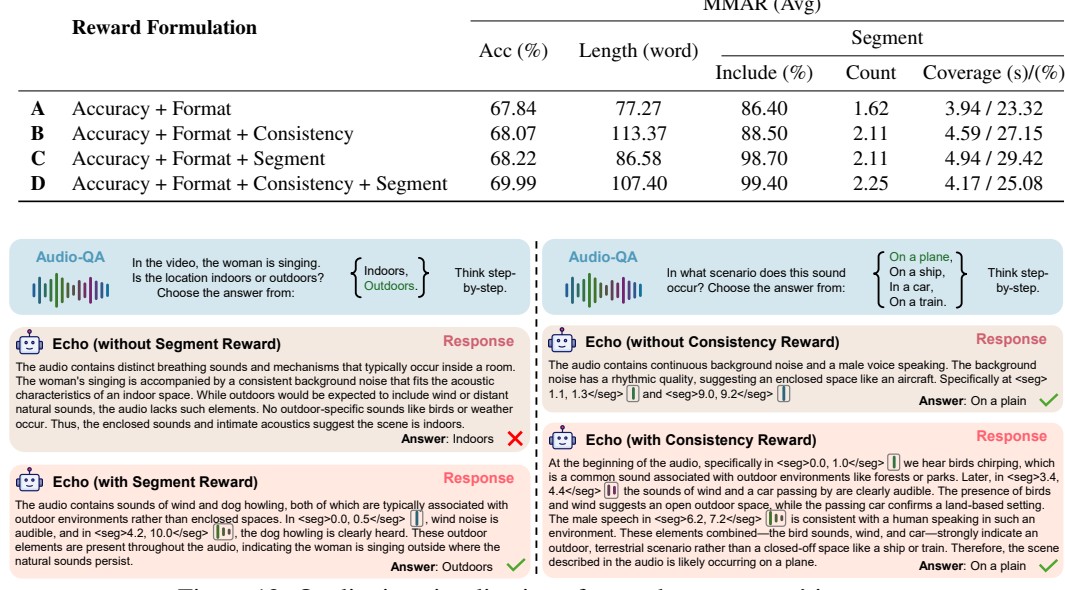

Figure 12: Qualitative visualization of reward components' impact.

Table 11 and Figure 12 examine the impact of different reward components during RL. In the right case of Figure 12, the model trained without the consistency reward exhibits unstable behavior after re-listening to audio segments, frequently skipping analytical steps and directly generating answers, leading to unreliable reasoning traces. In comparison, the model trained with the consistency reward produces coherent segment-wise analysis and derives conclusions in a structured, step-by-step

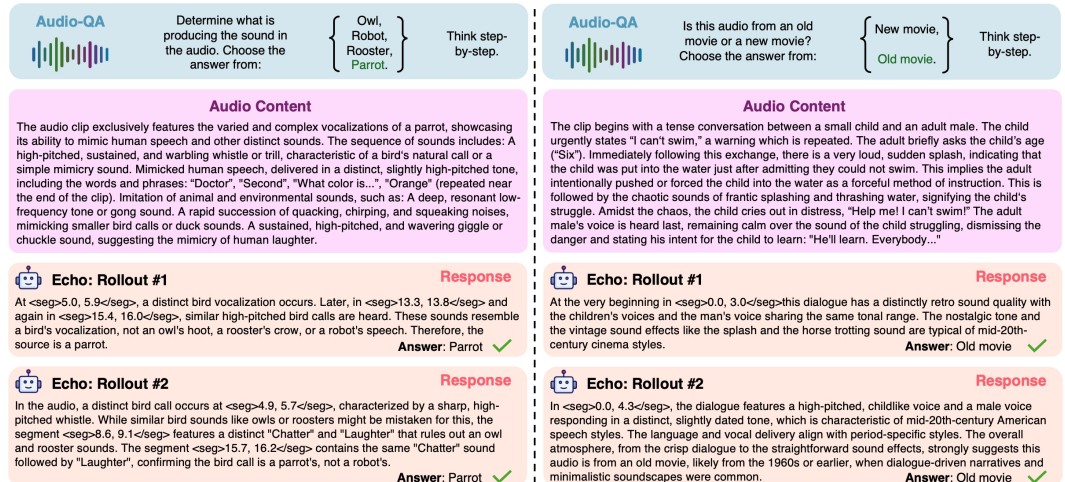

Figure 13: Audio-QA tasks typically permit multiple plausible reasoning paths.

manner. This effect is quantitatively reflected in Table 11 (lines **A→B**, **C→D**), where introducing the consistency reward consistently increases the average response length and the overall accuracy.

The segment reward, by contrast, primarily promotes greater reliance on audio segment references, as evidenced in transitions **A→C**, **B→D**. This encourages the integration of more comprehensive audio information and facilitates perception-grounded analysis. The advantage of the segment reward is also illustrated in the left example of Figure 12.

## J.2 THE DECISION AGAINST DIRECT PROCESS SUPERVISION

During the initial exploration, we do not seriously consider employing direct process supervision. This is primarily because our early experiments are conducted on the AVQA dataset, which provides only QA annotations and lacks the granular labels required for process-level supervision. Moreover, using an LLM-as-a-judge mechanism to compute rewards would substantially increase training costs due to API usage, and there is insufficient evidence at the time to confirm that LALMs can reliably evaluate reasoning processes or segment re-listening behavior.

As our focus shifts to the dataset we construct, the LLM-synthesized CoTs offer a potential reference for process supervision. We conduct small-scale experiments by separating high-quality CoTs from EAQA-SFT. The results show that, under identical RL data, reward signals based on textual semantic matching or segment IoU do not produce consistent performance gains and occasionally destabilize the RL process. We hypothesize that this occurs because many audio QA tasks permit a broad range of plausible reasoning paths, and imposing explicit constraints on these paths may unduly restrict the model's exploration.

As illustrated in Figure 13, the cues required to answer a question are not confined to a fixed combination of segments. In the left case, although the rollouts overlap partially, rollout #1 identifies the answer based on repeated high-pitched bird calls, whereas rollout #2 relies primarily on the parrot's characteristic chatter and laughter. In the right case, both rollouts re-listen to similar audio segments yet follow divergent reasoning paths: one focuses on retro sound quality, while the other attends to a speaker's dated tone and vocal delivery, and both ultimately lead to the correct answer.

Drawing on insights from thinking-with-image research, we ultimately adopt only a mild constraint (the segment reward), which encourages the model to invoke segment re-listening without being overly prescriptive about the reasoning trajectory. Empirically, this formulation has proven both effective and robust: it successfully increases the model's propensity to reference audio segments, and notably, we do not observe the anticipated reward hacking (e.g., superficial or random segment references) in practice. This iterative process leads to the final reward composition employed in our approach.

## K   GENERALIZABILITY TO LONG AUDIO

### K.1   DURATION-WISE ACCURACY

Table 12: Model accuracy comparison on MMAR for audio of different durations.

| Model | 1-10s 173 Samples | 11-20s 356 Samples | 21-30s 468 Samples | 31-60s 3 Samples |
|---|---|---|---|---|
| Base Model | 52.02 | 52.25 | 50.43 | 33.33 |
| Cold-Start Model | 56.07 | 61.80 | 51.28 | 66.67 |
| Echo | 67.63 | 70.22 | 67.31 | 100.00 |

Table 12 details the model accuracy across different audio lengths. Except for three overly lengthy cases, Echo demonstrates consistent proficiency in comprehending audio of varying durations, with a slight performance advantage in the 11-20 second range. Building upon the findings in Figure 6, these results confirm its ability to maintain accuracy when processing longer audio, confirming its generalizability.

The performance of the cold-start model reveals another noteworthy insight: although the audio used during SFT is limited to 10 seconds, the model exhibits its most pronounced performance improvement on 11–20 second audio. This observation suggests that, beyond audio-interleaved reasoning, the intermediate outcomes produced by audio-grounded reasoning also play a meaningful role in facilitating better handling of longer audio inputs.

Table 13: Model efficiency comparison on MMAR for audio of different durations.

| Model | 1-10s | | | 11-20s | | |
|---|---|---|---|---|---|---|
| | Length (word) | Latency (s) | Latency/Length | Length (word) | Latency (s) | Latency/Length |
| Base Model | 65.57 | 1.18 | 0.0180 | 64.32 | 1.09 | 0.0169 |
| Cold-Start Model | 107.12 | 1.82 | 0.0170 | 112.16 | 1.99 | 0.0177 |
| Echo | 96.39 | 1.99 | 0.0206 | 102.90 | 2.09 | 0.0203 |

| | 21-30s | | | All Samples | | |
|---|---|---|---|---|---|---|
| | Length (word) | Latency (s) | Latency/Length | Length (word) | Latency (s) | Latency/Length |
| Base Model | 71.31 | 1.26 | 0.0177 | 67.95 | 1.18 | 0.0174 |
| Cold-Start Model | 124.10 | 2.16 | 0.0174 | 117.06 | 2.04 | 0.0174 |
| Echo | 112.99 | 2.20 | 0.0195 | 107.40 | 2.12 | 0.0197 |

Table 13 summarizes the computational efficiency of different models across varying audio durations, measured by average response length, latency, and their ratio. The results show that, for all models, both response length and latency increase gradually as audio duration grows. For any fixed audio length, the cold-start model and Echo consistently produce substantially longer responses than the base model, which is consistent with the overall trends across all audio samples.

Given the conclusion from Table 3 where increased response length emerges as a natural consequence of SFT or RL, we further introduce the latency-to-length ratio to isolate the additional computational overhead specifically attributable to audio-interleaved reasoning. The analysis reveals that the segment-appending operation incurs only about a 13% increase in latency, and this overhead remains stable across different audio durations. These results indicate that the proposed reasoning strategy maintains acceptable computational efficiency, even when handling long audio.

### K.2   COVERAGE OF REVISITED SEGMENTS

Table 14: Proportion of re-listened segments intersecting with each time interval.

| Audio Duration | Intersect with [0-10s] | Intersect with [10.1-20s] | Intersect with [20.1-30s] |
|---|---|---|---|
| 0-10s (173 Samples) | 99.42 | - | - |
| 11-20s (356 Samples) | 97.75 | 24.16 | - |
| 21-30s (468 Samples) | 90.38 | 37.18 | 9.40 |
| 31-60s (3 Samples) | 100.00 | 66.67 | 0 |

Table 14 reports the proportion of re-listened segments that overlap with different time intervals, illustrating the temporal coverage of the Echo's re-listening behavior across audio of varying lengths. For instance, 97.75 in line 2 represents that 97.75% of Echo's responses to 11-20 second audio re-listen segments intersecting with [0-10s]. These results provide an intuitive view of the distribution of re-listened audio segments. Although segments beyond the 10-second mark are accessed less frequently than those within the first 10 seconds, the model still retrieves excerpts longer than 10 seconds—and even beyond 20 seconds—in a non-trivial fraction of cases. This fine-grained evidence further substantiates the model's generalization ability in segment retrieval and helps explain its sustained accuracy on 21–30 second audio.

## L LIMITATION ANALYSIS

### L.1 INFERIOR PERFORMANCE ON EASY VS. HARD TASKS

Table 15: Difficulty-wise accuracy comparison on MMAU-mini. Difficulty levels are provided as native metadata in the benchmark.

| Model | Easy | Medium | Hard |
|---|---|---|---|
| Base Model | 62.95 | 69.26 | 66.53 |
| Cold-Start Model | 64.29 | 75.37 | 69.49 |
| Echo | 75.00 | 84.07 | 77.12 |

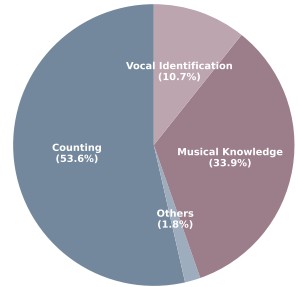

Figure 14: A task-wise error categorization for Echo within the easy subset.

Table 15 presents a closer examination of how the model's performance on tasks of varying difficulty evolves during training. A consistent trend emerges, with the model performing best on medium-difficulty tasks, followed by hard tasks, and least effectively on easy tasks. From our categorical analysis of error cases within the easy subset (Figure 14), we attribute this pattern to labeling bias in the benchmark's difficulty annotations. Most of the model's errors occur in areas such as counting, musical knowledge, and vocal identification. Although these tasks appear straightforward to human experts, they impose distinct challenges for the model:

**1). Counting** requires the accurate localization of all qualifying instances, often more than ten per task, which depends on reliable segment recognition. This ability is strongly influenced by the SFT process, yet our current SFT data construction does not sufficiently emphasize cases with highly frequent segment references within a single reasoning chain. **2). Musical knowledge** involves instruments, rhythm, chords, and other domain-specific concepts, and performance in these tasks largely reflects the coverage and quality of pre-training. As shown in Table 1 and Table 2, most LALMs underperform in music-related tasks compared to speech and sound, underscoring the need for more dedicated music-oriented pre-training data. **3). Vocal identification** is further complicated by the inherent variability of acoustic features, as speakers of similar gender and age may differ substantially in vocal characteristics. Addressing this challenge likely requires stronger task emphasis in both SFT and RL data design.

### L.2 FAILURE CASES

Figure 15 illustrates three representative types of errors made by Echo. In the left case, the model appropriately re-listens the audio segments but engages only in a superficial analysis of each, which prevents it from forming a logically coherent CoT that leads to the correct answer. In the top-right case, Echo demonstrates reasonable analysis in the early stages but subsequently confuses the number of words containing at least one stressed phoneme with the total number of stressed phonemes. This error is compounded by an inconsistency between the reasoning process and the final answer. A similar issue is also observed in the bottom-right case: the model correctly infers that the minor chord is Bm in the CoT, yet outputs F#m as the answer. These errors highlight the need for further improvements in both the comprehensiveness of reasoning and the consistency between reasoning and response.

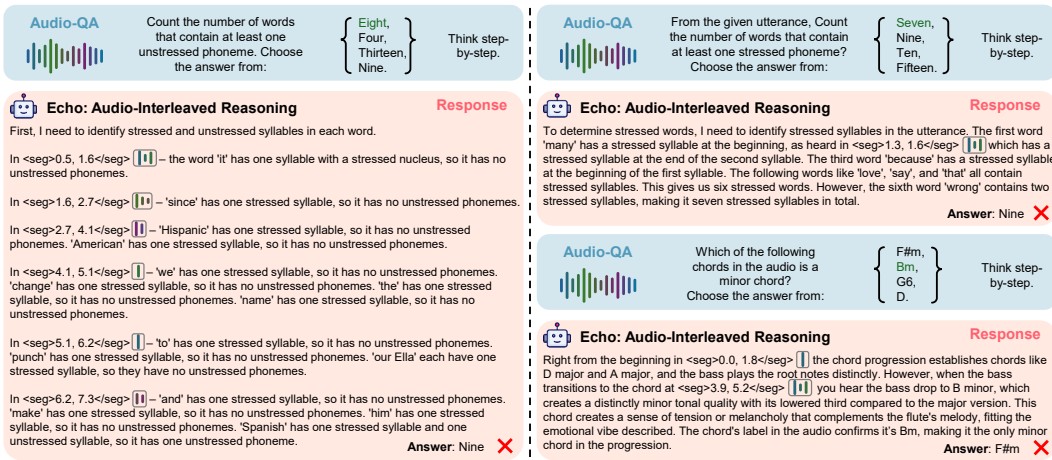

Figure 15: Three failure cases of Echo on MMAU-mini.

# M   MORE CASE VISUALIZATION

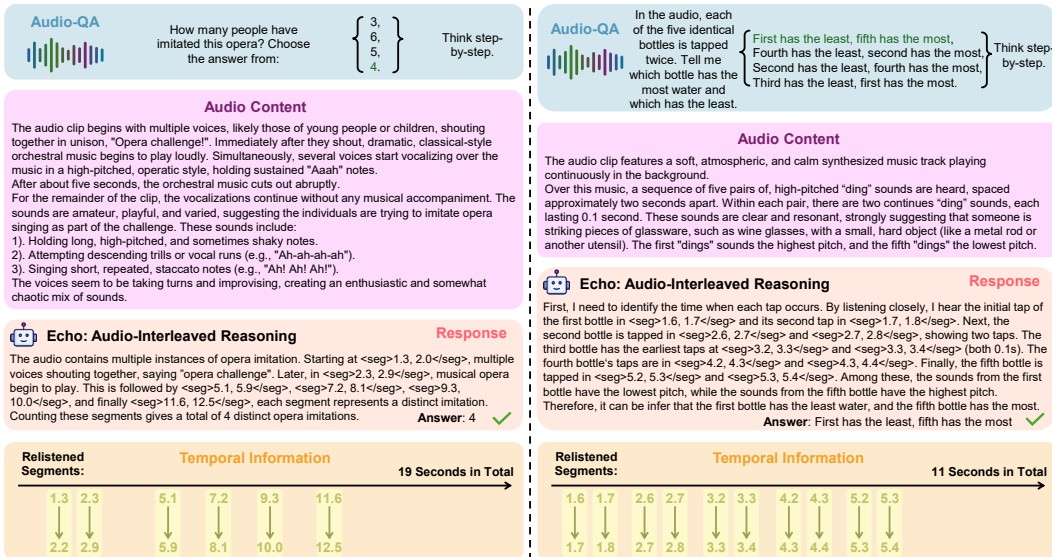

Figure 16: Case visualization on MMAR. The "audio content" is initially generated by Gemini-2.5-Pro (Comanici et al., 2025) and subsequently refined manually. Echo demonstrates expertise when handling continuous and frequent segment referring.

Figure 16 and Figure 17 present several case visualizations from MMAR, where Echo demonstrates distinctive capabilities in handling complex audio. Unlike previous visualizations, we provide an extended description of the audio content (generated via Gemini-2.5-Pro (Comanici et al., 2025) with human refinement) and annotate the relative positions of the referenced segments on a timeline.

In Figure 16, to count gunshots and assess the sharpness of tapping sounds on a water glass, Echo performs multiple consecutive segment re-listenings. Through accurate localization and coherent analysis, Echo successfully identifies nine gunshots and correctly infers the water level in the glass. In Figure 17, both audio clips are approximately 30 seconds long, significantly exceeding the 10-second segments encountered during SFT and RL. Moreover, their most informative regions are concentrated beyond the 10-second mark, presenting substantial challenges to the model. Despite this, Echo demonstrates the ability to reference segments beyond 10 seconds, even reaching the 29-second mark in the right-side case, and leverages these segments to form rational analyses leading to correct conclusions. This outcome highlights Echo's strong generalization in accessing and utilizing audio information across extended temporal contexts.

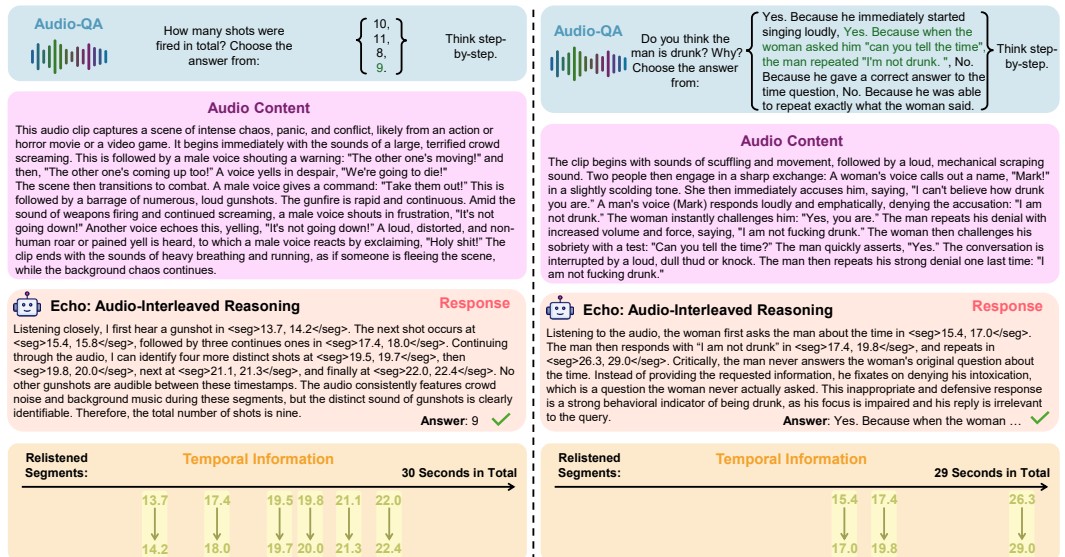

Figure 17: Case visualization on MMAR. Despite the duration of SFT data being constrained to 10 seconds, Echo can appropriately re-listen to segments after 10 seconds in demand.

# N    CLAIM OF LLM USAGE

LLMs, specifically DeepSeek-R1, serve as the primary contributors to the synthesis of both EAQA-SFT and EAQA-RL. Additionally, DeepSeek-R1 is also employed to polish the manuscript's writing, primarily by correcting grammatical errors and refining vocabulary usage.

