# OpenReview forum: "Echo: Towards Advanced Audio Comprehension via Audio-Interleaved Reasoning"
_ICLR.cc/2026/Conference — ICLR 2026 Poster_

### Official Review · Reviewer_jS6b · 2025-10-17

**Soundness:** 3
**Presentation:** 3
**Contribution:** 3
**Rating:** 8
**Confidence:** 3

**Summary:**

This paper introduces "Echo," a Large Audio Language Model training framework designed for audio comprehension. The novelty of the training method lies in the introduction of a new reasoning method called "audio-interleaved reasoning". Previous works encode audio only once, which the authors argue limit their ability to reason about subtle or complex audio details. In contrast, audio-interleaved reasoning allows the model to dynamically revisit and analyze specific audio segments during its reasoning process. The proposed training framework starts with supervised fine-tuning to locate important audio segments, followed by reinforcement learning to improve its ability to strategically revisit them.
The authors also presented a data generation pipeline that uses an LLM to synthesize Audio-QA data and corresponding audio-grounded CoTs for use in their RL and SFT training.
The authors demonstrated the performance of Echo by comparing it with various baselines on three benchmarks. The authors also performed ablation studies to analyze the effectiveness of different components of the framework.

**Strengths:**

Originality: The authors introduced a novel audio reasoning framework, "audio-interleaved reasoning", that differentiates this work from the standard "audio-conditioned text reasoning" format. The data generation pipeline is original as well.
Quality: The evaluation is comprehensive and the authors showed the advantage of the proposed framework over even proprietary systems. The ablation studies included analysis on different components of the training framework, further supporting the design decisions.
Clarity: The writing is easy to follow and the authors included many well-made figures to illustrate the framework and pipeline that are relatively complex.
Significance: The work has the potential to have significant impact on the field of audio understanding as it suggests a shift in paradigm that could inspire further works in audio-interleaved reasoning.

**Weaknesses:**

1. The model seems to be learning how to reason by imitating the reasoning trace of a separate reasoning LLM, Deepseek-R1, during the SFT phase. It's questionable whether the proposed model really learned how to reason in a more general and flexible sense.
2. As the authors have mentioned in the conclusion, the revisiting of the audio as implemented in this work is still rather straightforward.

**Questions:**

1. Have the authors tried any other reasoning LLMs besides deepseek-R1 to generate the training data? Or since the QA–CoT triplets are generated by deepseek-R1, have the authors tried using another reasoning LLM for the re-evaluation step so that the final triplets are not restricted to the reasoning pattern of deepseek-R1?
2. Did the authors tested the more advanced Gemini-2.5-Pro and Step-Audio-2 on MMAR? How do they compare?
3. Considering that the model has close to double the latency of the base model, has the authors investigated if the model is revisiting some audio segments when it's not necessary? Have the authors tried to minimize the segments revisited?

---

> ### Author Response · Authors · 2025-11-20
> **Response to Reviewer jS6b (1/3)**
>
> We sincerely appreciate your positive feedback and constructive advice! Below, we present detailed responses to the weaknesses (W) and questions (Q).
>
> >**W1. The model seems to be learning how to reason by imitating the reasoning trace of a separate reasoning LLM, Deepseek-R1, during the SFT phase. It's questionable whether the proposed model really learned how to reason in a more general and flexible sense.**
>
> Thank you for raising this important concern. In the SFT phase, our primary objective is for the model to learn how to reference salient audio segments via a well-defined format. To achieve this, SFT on CoTs that contain properly formatted audio references is a natural and practical solution. Since no such dataset is publicly available, we synthesize these traces using an LLM (DeepSeek-R1 in our implementation). We acknowledge that this setup inevitably causes the cold-start model to imitate the LLM's reasoning traces, as this is precisely what the SFT objective optimizes for. However, in our case, such imitation does not conflict with learning more generalizable reasoning capabilities. We support this claim with three empirical observations:
>
> (1). Before RL, we perform inference adaptation that transitions the model from audio-grounded reasoning to audio-interleaved reasoning. In the latter setting, the model must frequently process interleaved audio-text inputs, a format that is distinct from the single-audio reasoning used during SFT. Under this new format, the model, guided by outcome-supervised RL, still produces reasonable per-segment analysis and arrives at correct answers. **This indicates that the reasoning ability acquired during SFT generalizes effectively to interleaved  audio-text scenarios.**
>
> (2). During downstream evaluation, both MMAU and MMAR encompass a wide variety of tasks requiring diverse capabilities (e.g., MMAU covers 27 skills, MMAR includes 15). **Echo's overall strong performance across these benchmarks demonstrates a degree of task-wise generalization in reasoning.** Figure 5 of the manuscript further shows task-wise gains of the cold-start model over the base model across nearly all subskills, validating the isolated effect of SFT.
>
> |  | 1-10s |  | 11-20s |  | 21-30s |  |
> |---|:---:|:---:|:---:|:---:|:---:|:---:|
> |  | Acc | Length | Acc | Length | Acc | Length |
> | Base Model | 52.02 | 65.57 | 52.25 | 64.32 | 50.43 | 71.31 |
> | Cold-Start Model | 56.07 | 107.12 | 61.80 | 112.16 | 51.28 | 124.10 |
>
> (3). In duration-wise analysis, we evaluate the model on audio samples of varying durations, with summarized results presented in the table above (the complete version can be found in Appendix K of the updated manuscript). The results show that the cold-start model produces responses with average lengths distinct from those seen during SFT (87.5 words, presented in Table 4 of the updated manuscript). Moreover, despite all SFT CoTs being constrained under 10 seconds, the model produces longer and more detailed analyses for longer audio, while maintaining stable accuracy (on 11-20s audio). **The fact that the model adapts its reasoning length to the audio duration indicates temporal generalization beyond its training distribution.**
>
> Taken together, these findings suggest that the model acquires a degree of generalizable reasoning ability after SFT, rather than merely replicating the surface form of DeepSeek-R1's traces. At the same time, we acknowledge that SFT of this kind may introduce certain trade-offs in generalizability on non-audio tasks, which we consider outside the primary scope of this manuscript.
>
> >**W2. As the authors have mentioned in the conclusion, the revisiting of the audio as implemented in this work is still rather straightforward.**
>
> We agree that the current implementation is relatively straightforward. Since current LALMs do not inherently support audio-grounded reasoning, and there is a scarcity of training data containing rich audio-invoking CoT, our work primarily focuses on establishing a comprehensive training framework and a complementary data generation pipeline.
>
> By introducing these fundamental yet essential components, we aim to establish a baseline for audio-interleaved reasoning in LALMs. For this reason, we have not yet explored more customized or adaptive re-listening strategies. We fully acknowledge that there is ample room for such strategies, and we view this as an important direction for future research.

---

> > ### Author Response · Authors · 2025-11-20
> > **Response to Reviewer jS6b (2/3)**
> >
> > >**Q1. Have the authors tried any other reasoning LLMs besides deepseek-R1 to generate the training data? Or, since the QA–CoT triplets are generated by deepseek-R1, have the authors tried using another reasoning LLM for the re-evaluation step so that the final triplets are not restricted to the reasoning pattern of deepseek-R1?**
> >
> > | Synthesizer | Re-evaluator | # SFT Sample | # RL Sample | SFT CoT Length | MMAR | MMAU-mini |
> > |---|:---:|:---:|:---:|:---:|:---:|:---:|
> > | DeepSeek-R1 | DeepSeek-R1 | 75,862 | 21,900 | 87.5 words | 69.99 | 80.41 |
> > | Seed-1.6-Thinking | DeepSeek-R1 | 77,693 | 13,840 | 103.3 words | 68.51 | 78.78 |
> > | DeepSeek-R1 | GPT-5 | 54,523 | 15,325 | 82.2 words | 69.86 | 80.10 |
> >
> > Thank you for this constructive suggestion. We have conducted an ablation study to investigate the impact of using different LLMs (specifically, Seed-1.6-Thinking [1] and GPT-5 [2]) as the synthesizer and re-evaluator in our data generation pipeline. The results are summarized in the table above.
> >
> > To ensure a fair evaluation, we maintain the identical training configuration across all SFT and RL experiments as before. Although substituting either the synthesizer or the re-evaluator leads to variations in sample numbers and CoT lengths, the downstream model performance remains largely consistent, which validates the stability and robustness of the proposed data generation pipeline. Notably, both variants yield a substantial reduction in the amount of RL data, yet this reduction leads to only marginal performance drops. This pattern suggests a potential trade-off between data quality and quantity: stricter filtering may increase the overall reliability of the retained samples, thereby enabling more data-efficient optimization.
> >
> > These results can be found in Appendix F.3 of the updated manuscript.
> >
> > **References**
> >
> > [1] Seed1.6 Tech Introduction. 2025. \
> > [2] GPT-5 System Card. 2025.
> >
> > >**Q2. Did the authors test the more advanced Gemini-2.5-Pro and Step-Audio-2 on MMAR? How do they compare?**
> >
> > We have evaluated Gemini-2.5-Pro on MMAR, and the results have been included in the updated manuscript (Table 1, Line 284). Gemini-2.5-Pro achieves a macro-averaged accuracy of 70.03% on MMAR, which slightly surpasses Echo's performance. We have accordingly adjusted our analysis to provide a more accurate positioning of our model's capabilities.
> >
> > Regarding Step-Audio-2, to the best of our knowledge, its API does not currently support custom audio understanding. Furthermore, its technical report does not include evaluations on MMAR, nor have we found any published work reporting its performance on this benchmark. Therefore, we are unfortunately unable to provide a direct comparison at this time. That said, we would be glad to incorporate such results in the manuscript if they become available.

---

> > > ### Author Response · Authors · 2025-11-20
> > > **Response to Reviewer jS6b (3/3)**
> > >
> > > >**Q3. Considering that the model has close to double the latency of the base model, has the authors investigated if the model is revisiting some audio segments when it's not necessary? Have the authors tried to minimize the segments revisited?**
> > >
> > > | Model | SFT Data | RL Data | Reasoning Format | Length (word) | Latency (s) | Latency/Length |
> > > |---|:---:|:---:|:---:|:---:|:---:|:---:|
> > > | Base Model | - | - | Audio-Conditioned Text | 67.95 | 1.18 | 0.0174 |
> > > | Cold-Start Model | EAQA-SFT | - | Audio-Grounded | 117.06 | 2.04 | 0.0174 |
> > > | Unadapted RL Model | EAQA-SFT | EAQA-RL | Audio-Grounded | 98.12 | 1.97 | 0.0201 |
> > > | Cold Start Model | EAQA-SFT | - | Audio-Interleaved | 101.73 | 2.16 | 0.0212 |
> > > | Direct RL Model | - | EAQA-RL | Audio-Conditioned Text | 104.42 | 2.06 | 0.0197 |
> > > | Direct RL Model | - | AVQA | Audio-Conditioned Text | 111.04 | 2.11 | 0.0190 |
> > > | Echo-AVQA | EAQA-SFT | AVQA | Audio-Interleaved | 120.18 | 2.37 | 0.0197 |
> > > | Echo | EAQA-SFT | EAQA-RL | Audio-Interleaved | 107.40 | 2.12 | 0.0197 |
> > >
> > > We acknowledge that Echo has close to double the latency of the base model. However, our ablation studies above (summarized from Table 3 in the manuscript) indicate that this increase is not solely attributable to audio-interleaved reasoning. The results show that both SFT and RL contribute to the latency increase, independently of the reasoning format or the specific RL dataset used. To isolate the additional computational overhead specifically attributable to audio-interleaved reasoning, we further introduce the latency-to-length ratio. The analysis reveals that the segment-appending operation incurs only about a 13\% (0.0197/0.0174) increase in this ratio, indicating that **the proposed reasoning strategy itself maintains acceptable computational efficiency**.
> > >
> > > Regarding the concern about potential unnecessary segment revisiting, we kindly refer you to **General Response 2**. This response window includes a comprehensive evaluation of the model's segment localization and per-segment analysis. The results demonstrate that Echo exhibits high proficiency in accurately locating audio segments and performs the corresponding analysis as intended. Together with the results in Figure 4 of the manuscript, where the overlap between visited segments remains relatively low (<15%), these findings suggest that **unnecessary revisiting is likely limited and that the model predominantly triggers re-listening only when it is contextually motivated**.
> > >
> > > Concerning efforts to minimize revisited segments, we have not explicitly experimented with reducing the number of segment calls. In the current design, the only constraint on segment re-listening is a segment reward that encourages the model to make at least one reference, a choice guided by empirical observations (Since Reviewer rXCQ shares a similar concern, these responses partially overlap). During exploration, we experiment with more explicit forms of process supervision, such as rewards based on textual semantic matching or segment IoU. However, we find that these do not yield consistent performance gains and occasionally destabilize the RL training process. We hypothesize that this occurs because many audio QA tasks permit a broad range of plausible reasoning paths, and imposing explicit constraints on these paths may unduly restrict the model's exploration. Drawing on insights from "thinking with image" research, we adopt the current formulation, which has proven both effective and robust in practice.
> > >
> > > However, we fully agree that developing methods to further optimize or constrain segment usage, such as minimizing unnecessary revisits, is an interesting and meaningful direction, and we consider it a promising avenue for future work.

---

### Official Review · Reviewer_H3Dr · 2025-11-01

**Soundness:** 3
**Presentation:** 3
**Contribution:** 3
**Rating:** 4
**Confidence:** 5

**Summary:**

The paper introduces Echo, an LALM designed to perform audio-interleaved reasoning. For this, the authors propose a novel framework that allows the model to actively revisit audio segments during reasoning, rather than relying on a single static audio encoding. Inspired by human cognitive processes, the approach aims to overcome the information bottleneck of prior audio-conditioned text reasoning. The authors train Echo using a 2-step framework, which includes an SFT stage and an RL stage. They also propose their own data generation pipeline that combines Qwen2.5-Omni captions and DeepSeek-R1 prompting (as a result, they propose 2 new datasets). The proposed model achieves significant improvements across audio understanding and reasoning benchmarks.

**Strengths:**

- The paper tries something new, and I am glad that it explores the aspect that is missing in audio.
- Strong empirical gains on diverse and challenging benchmarks.
- Thorough ablation and analytical studies validating each training stage.
- Introduction of high-quality synthetic Audio-QA datasets with CoTs.
- Efficient implementation and transparent reproducibility commitments. (I hope the model is open-sourced)

**Weaknesses:**

- (Minor) Grounding for reasoning is not new and has been extensively explored in vision (images and videos). Here are some examples:
- I strongly disagree with the claim that previous works suffer from “contextualizing audio content through a one-time encoding, which introduces a critical information bottleneck” — I understand what the authors are trying to convey, but the problem statement has not been framed well. This is a grounding issue that the authors are trying to solve. The other way to frame it would be: “Current methods do not ground to the audio while reasoning but Echo does” (I have cited some papers below that does grounding in vision and authors can check how they claim). — A good example is the fact that for prior methods, during the reasoning process, when each token is generated during or after the reasoning process, the model constantly focuses and re-focuses on separate parts of the audio for the next token (through attention maps in self-attention), which is also a form of “attending to the audio multiple times” — as claimed by the authors which Echo explicitly does. There are some papers on “playback” which better fits your narrative [].
- The data generation pipeline heavily relies on synthetic annotations (DeepSeek-R1, Qwen2.5), which may introduce model-specific biases. I do not see any verification methods. Additionally, I have personally gone through several of AudioSet-Strong and some of MusicBench timestamps. They are very noisy. I wonder how good the generated dataset is.
- (Minor) I am shocked by the 69.99 on MMAR. Did the authors evaluate with the entire thinking chain and the answer or just the answer?
- (Minor) A human evaluation of the thinking chain would have been good.

**Questions:**

- I have asked some questions about the weakness.
- I may be confused, but how does interleaving happen during test time? Is a tool called for cutting the audio? Where is that information on the paper?
- Does the reward include rewards for the thought? Since the entire audio is also part of the context, how do we know if the final answer comes from the interleaved thought or just the entire audio? I do not see a reward for if the useful segments were interleaved?

---

> ### Author Response · Authors · 2025-11-20
> **Response to Reviewer H3Dr (1/5)**
>
> We sincerely appreciate your insightful critiques and constructive suggestions! Below, we present detailed responses to the weaknesses (W) and questions (Q).
>
> >**W1. (Minor) Grounding for reasoning is not new and has been extensively explored in vision (images and videos). Here are some examples.**
>
> We fully acknowledge that grounding for reasoning has indeed been extensively explored recently in the vision domain, which serves as one of the key inspirations for our proposal of audio-interleaved reasoning. The temporal nature of audio allows for natural grounding and segmentation; however, the seamless interleaving of the raw audio signal into the reasoning process remains underexplored, as you recognize in our strengths. As we state in the related work of the manuscript, we hope to introduce audio-interleaved reasoning to catalyze a similar evolution in LALMs from "thinking about audio" to "thinking with audio".
>
> On the other hand, compared to the rich body of work in LVLMs, research on LALMs is relatively scarce, which introduces additional implementation challenges. Many advanced LVLMs (e.g., Qwen2.5-VL) possess innate grounded reasoning capabilities, and the LLM-as-a-judge paradigm is supported by more empirical evidence. These advantages allow most "thinking with image" works to focus more on pure RL processes, including reward design and loss weighting. In contrast, current advanced LALMs (e.g., Qwen2.5-Omni) inherently lack such grounded reasoning abilities, necessitating the construction of a complete data generation pipeline and the adaptation of existing SFT and RL training frameworks. By systematically addressing these gaps, it is our hope that this work can establish a solid foundation for "thinking with audio" and inspire future research in this emerging direction.

---

> > ### Author Response · Authors · 2025-11-20
> > **Response to Reviewer H3Dr (2/5)**
> >
> > >**W2. I strongly disagree with the claim that previous works suffer from “contextualizing audio content through a one-time encoding, which introduces a critical information bottleneck” — I understand what the authors are trying to convey, but the problem statement has not been framed well. This is a grounding issue that the authors are trying to solve. The other way to frame it would be: “Current methods do not ground to the audio while reasoning, but Echo does” (I have cited some papers below that does grounding in vision and authors can check how they claim). — A good example is the fact that for prior methods, during the reasoning process, when each token is generated during or after the reasoning process, the model constantly focuses and re-focuses on separate parts of the audio for the next token (through attention maps in self-attention), which is also a form of “attending to the audio multiple times” — as claimed by the authors which Echo explicitly does. There are some papers on “playback” which better fit your narrative [].**
> >
> > Thank you for the constructive feedback. Upon carefully reviewing the framing of the motivation in our initial manuscript and referring to relevant "thinking with image" literature, we recognize the validity of your point and agree that our original presentation could have been clearer.
> >
> > In the original manuscript, we inaccurately describe Echo's behavior of invoking audio as "revisiting" and presented this as an advantage absent in prior works. As you rightly pointed out, existing methods also *"constantly focus and re-focus on separate parts of the audio for the next token"* during the reasoning process. This indicates that "revisit" is not unique to audio-interleaved reasoning, making our initial characterization inappropriate.
> >
> > On the other hand, you express disagreement with our statement that "*previous works suffer from contextualizing audio content through a one-time encoding, which introduces a critical information bottleneck*", and suggest that the issue might instead be framed as a grounding problem. With great respect, we would like to provide a different perspective.
> >
> > After carefully reviewing relevant prior work (the citations you provided unfortunately did not render properly on OpenReview), we find that existing grounded reasoning methods in vision can be grouped into two categories. The first category, exemplified by GRIT [1], injects bounding-box coordinates into the reasoning trace, rather than raw pixel inputs. We believe this line of work aligns more closely with the grounding issue you highlight. The second category, exemplified by DeepEye [2], extends the first category by modifying the reasoning mechanism to allow direct insertion of pixel tokens. **Our proposed audio-interleaved reasoning is conceptually analogous to this latter category, taking an additional step beyond grounded reasoning**.
> >
> > Within this context, although your alternative framing — "*Current methods do not ground to the audio while reasoning, but Echo does*" — is correct, it does not fully capture all advantages. **While grounding through coordinate insertion, like GRIT, indeed guides models to refocus on visual tokens, the visual inputs accessible throughout the reasoning are still bottlenecked by the original, one-time visual encoding.** This aligns with what we refer to as the information bottleneck: *the difficulty of preserving subtle audio details and reconstructing them from heavily compressed embeddings (Lines 52–53 of the updated manuscript, consistent with the original version)*.
> >
> > In contrast, incorporating pixel tokens (as in DeepEye) or audio tokens (as in Echo) introduces additional multimodal context into the reasoning process, potentially capturing details overlooked by the initial encoding and thereby overcoming this bottleneck. Our use of this phrasing is also inspired by a survey [3], which similarly highlights this bottleneck as a key motivation for "thinking with image".
> >
> > Based on this analysis, we have carefully revised the inappropriate characterization in the manuscript and replaced "revisit" with "re-listen", while retaining the "information bottleneck" formulation. We greatly value this opportunity to discuss these nuances with you and look forward to addressing any remaining concerns you may have.
> >
> > **References**
> >
> > [1]. GRIT: Teaching MLLMs to Think with Images. NeurIPS 2025. \
> > [2]. DeepEyes: Incentivizing" Thinking with Images" via Reinforcement Learning. arXiv 2025. \
> > [3]. Thinking with Images for Multimodal Reasoning: Foundations, Methods, and Future Frontiers. arXiv 2025.

---

> > > ### Author Response · Authors · 2025-11-20
> > > **Response to Reviewer H3Dr (3/5)**
> > >
> > > >**W3. The data generation pipeline heavily relies on synthetic annotations (DeepSeek-R1, Qwen2.5), which may introduce model-specific biases. I do not see any verification methods. Additionally, I have personally gone through several of AudioSet-Strong and some of MusicBench timestamps. They are very noisy. I wonder how good the generated dataset is.**
> > >
> > > Thank you for raising these important concerns. Regarding model-specific biases, we fully acknowledge the validity of this and therefore explicitly highlight and discuss this limitation in the Ethics Statement of our updated manuscript. The content reads: "However, we note that the use of LLM (DeepSeek-R1) in the data generation process may inadvertently propagate biases inherent in its underlying training data. Specifically, as revealed by [1], DeepSeek-R1 exhibits systematic "rumination", characterized by repeatedly revisiting similar problem framings rather than exploring diverse reasoning trajectories. The same study also highlights that DeepSeek-R1 may generate harmful or unsafe reasoning traces, which could, in turn, be unintentionally distilled into downstream models during training."
> > >
> > > Furthermore, we have conducted additional ablation experiments to systematically evaluate the impact of using different LLMs (specifically, Seed-1.6-Thinking [2] and GPT-5 [3]) as both synthesizers and re-evaluators in the pipeline, with the results summarized in the table below.
> > >
> > > | Synthesizer | Re-evaluator | # SFT Sample | # RL Sample | SFT CoT Length | MMAR | MMAU-mini |
> > > |---|:---:|:---:|:---:|:---:|:---:|:---:|
> > > | DeepSeek-R1 | DeepSeek-R1 | 75,862 | 21,900 | 87.5 words | 69.99 | 80.41 |
> > > | Seed-1.6-Thinking | DeepSeek-R1 | 77,693 | 13,840 | 103.3 words | 68.51 | 78.78 |
> > > | DeepSeek-R1 | GPT-5 | 54,523 | 15,325 | 82.2 words | 69.86 | 80.10 |
> > >
> > > To ensure a fair evaluation, we maintain the identical training configuration across all SFT and RL experiments as before. Although substituting either the synthesizer or the re-evaluator leads to variations in sample numbers and CoT lengths, the downstream model performance remains largely consistent, which **validates the stability and robustness of the proposed data generation pipeline to model-specific biases**. These results can be found in Appendix F.3 of the updated manuscript.
> > >
> > > Regarding the mitigation of noise from timestamps, our annotation pipeline achieves this through two complementary mechanisms (Since Reviewer rXCQ shares a similar concern, these responses overlap).
> > >
> > > First, during the synthesis of QA-CoT triplets, the LLM (DeepSeek-R1) is provided with audio information from multiple sources, including temporal metadata and predictions from Qwen2.5-Omni. **These sources implicitly cross-validate one another, allowing the LLM to assess the relative reliability of different cues**. This mechanism promotes a more selective use of temporal annotations, thereby mitigating the propagation of noise into synthesized data.
> > >
> > > Second, **the LLM performs an additional round of re-evaluation to further improve the quality of the synthesized training data**. In this stage, temporal metadata, audio information extracted from Qwen2.5-Omni, and the previously synthesized QA-CoT triplets are jointly provided to the LLM, allowing it to reassess each sample with full contextual evidence. Through explicit prompting, the LLM is instructed to verify answer plausibility, identify ambiguous choices, and detect unsupported or hallucinated CoT steps. As a result, the annotation noise and hallucinated reasoning are further suppressed in the final training sets. **Two representative examples are illustrated in Appendix F.2 of the updated manuscript.**

---

> > > > ### Author Response · Authors · 2025-11-20
> > > > **Response to Reviewer H3Dr (4/5)**
> > > >
> > > > >**Continuing W3. The data generation pipeline heavily relies on synthetic annotations (DeepSeek-R1, Qwen2.5), which may introduce model-specific biases. I do not see any verification methods. Additionally, I have personally gone through several of AudioSet-Strong and some of MusicBench timestamps. They are very noisy. I wonder how good the generated dataset is.**
> > > >
> > > > **Furthermore, these two mechanisms can also be validated through quantitative evidence.** The table below specifically substantiates the effect of the first mechanism (implicit cross-validation) by comparing the impact of different audio extractors on data quality and downstream performance:
> > > >
> > > > | Audio Extractor | MMAR (Acc%) | MMAU-mini (Acc%) |
> > > > |---|:---:|:---:|
> > > > | Qwen2.5-Omni | 69.99 | 80.41 |
> > > > | Qwen2-Audio | 63.91 | 75.41 |
> > > >
> > > > The results show that using a more capable audio extractor (to extract audio description, speech transcription, and musical analysis) leads to significantly higher model accuracy on both benchmarks. **This demonstrates how enhanced audio information enables more effective implicit cross-validation during triplet synthesis, thereby reducing noise propagated from temporal metadata and improving the overall quality of the generated data.**
> > > >
> > > > The impact of the second mechanism (explicit re-evaluation) is quantitatively validated in the table below:
> > > >
> > > > | SFT Data | RL Data | MMAR (Acc%) | MMAU-mini (Acc%) |
> > > > |---|---|:---:|:---:|
> > > > | 76k Samples Re-evaluated as High Quality | 22k Samples Re-evaluated as High Quality | 69.99 | 80.41 |
> > > > | 111k Unfiltered Samples (Expect for RL Data) | 22k Samples Re-evaluated as High Quality | 64.67 | 76.19 |
> > > > | 76k Samples Re-evaluated as High Quality | 22k Samples Re-evaluated as Low Quality | 67.03 | 77.40 |
> > > >
> > > > From the results, the model achieves optimal results when both SFT and RL data consist solely of high-quality samples (first row). Introducing low-quality samples into SFT data causes a clear performance drop (second row), confirming that re-evaluation effectively reduces CoT noise and hallucinations. When RL data is replaced with low-quality samples (a cost-efficient alternative to scaling RL), performance also declines (third row). **This highlights the effectiveness of re-evaluation in mitigating data noise.** The complete versions of these two experiments are provided in Appendix F.1&F.2 in the updated manuscript.
> > > >
> > > > **References**
> > > >
> > > > [1] DeepSeek-R1 Thoughtology: Let's Think about LLM Reasoning. arXiv 2025. \
> > > > [2] Seed1.6 Tech Introduction. 2025. \
> > > > [3] GPT-5 System Card. 2025.
> > > >
> > > > >**W4. (Minor) I am shocked by the 69.99 on MMAR. Did the authors evaluate with the entire thinking chain and the answer or just the answer?**
> > > >
> > > > In evaluation, we consider an answer correct if and only if the content enclosed within the [answer] (<> can not be correctly shown on OpenReview) tags matches exactly with the ground-truth, ignoring case and special characters. This approach guarantees that each model response is mapped to at most one answer, preventing any unfair comparisons. We have clarified this in Lines 317-320 of the updated manuscript to eliminate potential ambiguity.
> > > >
> > > > >**W5. (Minor) A human evaluation of the thinking chain would have been good.**
> > > >
> > > > We kindly refer you to our **General Response 2**. In this response window, we have supplemented the evaluation of the reasoning chain by not only comparing the model's segment localization performance against **human annotations**, but also adopting an LLM-as-a-judge approach to assess the relevance of its per-segment analysis to the audio content, as well as its contribution to the final conclusion. The results demonstrate that Echo exhibits high proficiency in accurately locating audio segments and performs the corresponding analysis as intended. These results can also be found in Appendix I of the updated manuscript.

---

> > > > > ### Author Response · Authors · 2025-11-20
> > > > > **Response to Reviewer H3Dr (5/5)**
> > > > >
> > > > > >**Q1. I may be confused, but how does interleaving happen during test time? Is a tool called for cutting the audio? Where is that information on the paper?**
> > > > >
> > > > > During inference, Echo performs audio-interleaved reasoning following the identical adaptation detailed in Section 3.1 of the manuscript. Specifically, when processing an input containing both audio and a question, the model generates text until it detects a special segmentation tag pair. This tag pair contains start and end timestamps, which are used to extract the corresponding segment from the original audio. **The extraction operation does not necessarily require external tools, which can be executed by directly clipping the raw audio array based on the timestamps and the known sampling rate.** The clipped audio segment is then combined with the current text output to form an augmented input sequence. This enriched sequence is fed back into the model to continue the reasoning process. This cycle of generating text, extracting audio segments based on the generated timestamps, and augmenting the input repeats iteratively until the model produces an end-of-sequence token, completing the inference process.
> > > > >
> > > > > We have clarified this in Lines 317-318 of the updated manuscript.
> > > > >
> > > > > >**Q2. Does the reward include rewards for the thought? Since the entire audio is also part of the context, how do we know if the final answer comes from the interleaved thought or just the entire audio? I do not see a reward for if the useful segments were interleaved?**
> > > > >
> > > > > We opt not to use rewards for thought based on empirical observations (Since Reviewer rXCQ shares a similar concern, these responses overlap). During exploration, we experiment with more explicit forms of process supervision, such as rewards based on textual semantic matching or segment IoU. However, **we find that these do not yield consistent performance gains and occasionally destabilize the RL training process.** We hypothesize that this occurs because many audio QA tasks permit a broad range of plausible reasoning paths, and imposing explicit constraints on these paths may unduly restrict the model's exploration.
> > > > >
> > > > > Drawing on insights from "thinking with image" research, we ultimately adopt only a mild constraint (the segment reward), which encourages the model to invoke segment re-listening without being overly prescriptive about the reasoning trajectory. **Empirically, this formulation has proven both effective and robust**: it successfully increases the model's propensity to reference audio segments (as validated in Table 11 of the updated manuscript), and notably, we do not observe the anticipated reward hacking in practice. This iterative process leads to the final reward composition employed in our approach.
> > > > >
> > > > > Furthermore, **the feasibility of this reward design is strongly corroborated by the comprehensive quantitative results on segment localization and per-segment analysis (as in General Response 2)**. This empirical finding demonstrates that the model, under the current reward scheme, not only revisits segments but does so in a manner that is accurate and beneficial for the reasoning process, thereby validating our design choice.

---

### Official Review · Reviewer_Dkpf · 2025-11-01

**Soundness:** 3
**Presentation:** 3
**Contribution:** 3
**Rating:** 4
**Confidence:** 4

**Summary:**

The paper presents an audio-interleaving reasoning method to improve reasoning abilities in LALMs. The method revisits audio segments during the thinking process for more effective localization and analysis. In order to train this ability the paper proposes a data generation method that utilizes rich temporal annotations in AudioSet-SL and MusicBench. The paper proposes a SFT+RL framework with a custom reward that considers formats, semantic continuity, accuracy, and segment revisiting. Experiments reveal that the proposed ECHO method improves on MMAU and MMAR, and ablation studies demonstrate the effect of each part of the method and the detailed improvements on different aspects.

**Strengths:**

- The writing of the paper is very clear.
- The segment revisiting approach is novel in the LALM domain and it makes a lot of sense to expect it to improve reasoning quality, so it is a good intuition.
- The data generation and annotation pipeline is clear and easy for reproducing, thus benefiting the community to build upon this work.
- The main results indicate that applying ECHO has consistent better results on MMAU and MMAR. In addition, ablation studies clearly show the benefits of audio-interleaving RL and appending segments, and the improvements are consistent accross different subsets of MMAU.
- There are extra ablation experiments and analysis including error analysis in the appendix, making the analysis of the proposed method very extensive.

**Weaknesses:**

I think the main weakness of this paper is that current experiments do not prove the quality (such as accuracy) of the segment selection, which is a fundamental basis of this paper. In data generation, the paper uses AudioSet-SL and MusicBench. It is known that AudioSet-SL annotations are very noisy, and this could impose noisy training data during data generation. In RL training, the "format" and "consist" rewards are mostly for formats; the "acc" reward is for the final prediction; the "seg" reward encourages revisiting the segment but it does not necessarily guarantee whether the revisiting is reliable -- that is, the model might just hallucinate during this step and still gets points. In evaluation, the paper also does not evaluate whether the reasoning process correctly extracts useful segments and conducts reasonable thinking for the segment. In my view, there are two ways to fix this problem. First, explicitly parse the segments and per-segment analysis from the outputs and query a good LALM for verification. Second, one can construct a real or synthetic testset with known, accurate time annotations and see if the model can output exactly the same segments and desirable analysis.

**Questions:**

- Does the framework apply to complex cases where the model has to listen to several distinct segments at the same time?
- The segment appending operation seems to impact the model's performance on long audio understanding. How to address this efficiency issue?
- How do you filter inaccurate training samples in Fig 3 especially if the original timestamps are very inaccurate (such as some inaccurately annotated samples in AudioSet-SL)?
- How does the proposed reward make sure the per-segment reasoning is correct, not hallucination?
- Can you evaluate the segmenting accuracy and per-segment analysis quality in order to further justify your framework?

---

> ### Author Response · Authors · 2025-11-20
> **Response to Reviewer Dkpf (1/3)**
>
> We sincerely appreciate your thorough review and constructive suggestions! Below, we present detailed responses to the weaknesses (W) and questions (Q).
>
> >**W1. I think the main weakness of this paper is that current experiments do not prove the quality (such as accuracy) of segment selection, which is a fundamental basis of this paper.**
>
>  Inspired by your suggestions in **W4**, we have conducted a comprehensive quantitative analysis of segment selection and per-segment analysis, which is now presented in **General Response 2**. **The results demonstrate that Echo exhibits high proficiency in accurately locating audio segments and performs the corresponding analysis as intended.** These results can also be found in Appendix I of the updated manuscript.
>
> >**W2. In data generation, the paper uses AudioSet-SL and MusicBench. It is known that AudioSet-SL annotations are very noisy, and this could impose noisy training data during data generation.**
>
> We agree with you that our adopted temporal metadata contains certain noise, and our data annotation pipeline reduces such noise through two complementary mechanisms (Since Reviewer rXCQ shares a similar concern, these responses overlap).
>
> First, during the synthesis of QA-CoT triplets, the LLM (DeepSeek-R1) is provided with audio information from multiple sources, including temporal metadata and predictions from Qwen2.5-Omni. **These sources implicitly cross-validate one another, allowing the LLM to assess the relative reliability of different cues**. This mechanism promotes a more selective use of temporal annotations, thereby mitigating the propagation of noise into synthesized data.
>
> Second, **the LLM performs an additional round of re-evaluation to further improve the quality of the synthesized training data**. In this stage, temporal metadata, audio information extracted from Qwen2.5-Omni, and the previously synthesized QA-CoT triplets are jointly provided to the LLM, allowing it to reassess each sample with full contextual evidence. Through explicit prompting, the LLM is instructed to verify answer plausibility, identify ambiguous choices, and detect unsupported or hallucinated CoT steps. As a result, the annotation noise and hallucinated reasoning are further suppressed in the final training sets. **Two representative examples are illustrated in Appendix F.2 of the updated manuscript.**
>
> **Furthermore, these two mechanisms can also be validated through quantitative evidence.** The table below specifically substantiates the effect of the first mechanism (implicit cross-validation) by comparing the impact of different audio extractors on data quality and downstream performance:
>
> | Audio Extractor | MMAR (Acc%) | MMAU-mini (Acc%) |
> |---|:---:|:---:|
> | Qwen2.5-Omni | 69.99 | 80.41 |
> | Qwen2-Audio | 63.91 | 75.41 |
>
> The results show that using a more capable audio extractor (to extract audio description, speech transcription, and musical analysis) leads to significantly higher model accuracy on both benchmarks. **This demonstrates how enhanced audio information enables more effective implicit cross-validation during triplet synthesis, thereby reducing noise propagated from temporal metadata and improving the overall quality of the generated data.**
>
> The impact of the second mechanism (explicit re-evaluation) is quantitatively validated in the table below:
>
> | SFT Data | RL Data | MMAR (Acc%) | MMAU-mini (Acc%) |
> |---|---|:---:|:---:|
> | 76k Samples Re-evaluated as High Quality | 22k Samples Re-evaluated as High Quality | 69.99 | 80.41 |
> | 111k Unfiltered Samples (Expect for RL Data) | 22k Samples Re-evaluated as High Quality | 64.67 | 76.19 |
> | 76k Samples Re-evaluated as High Quality | 22k Samples Re-evaluated as Low Quality | 67.03 | 77.40 |
>
> From the results, the model achieves optimal results when both SFT and RL data consist solely of high-quality samples (first row). Introducing low-quality samples into SFT data causes a clear performance drop (second row), confirming that re-evaluation effectively reduces CoT noise and hallucinations. When RL data is replaced with low-quality samples (a cost-efficient alternative to scaling RL), performance also declines (third row). **This highlights the effectiveness of re-evaluation in mitigating data noise.** The complete versions of these two experiments are provided in Appendix F.1&F.2 in the updated manuscript.

---

> > ### Author Response · Authors · 2025-11-20
> > **Response to Reviewer Dkpf (2/3)**
> >
> > >**W3. In RL training, the "format" and "consist" rewards are mostly for formats; the "acc" reward is for the final prediction; the "seg" reward encourages revisiting the segment but it does not necessarily guarantee whether the revisiting is reliable -- that is, the model might just hallucinate during this step and still gets points.**
> >
> > Our choice of reward implementation is driven primarily by empirical observations (Since Reviewer rXCQ shares a similar concern, these responses overlap). During exploration, we experiment with more explicit forms of process supervision, such as rewards based on textual semantic matching or segment IoU. However, **we find that these do not yield consistent performance gains and occasionally destabilize the RL training process.** We hypothesize that this occurs because many audio QA tasks permit a broad range of plausible reasoning paths, and imposing explicit constraints on these paths may unduly restrict the model's exploration.
> >
> > Drawing on insights from "thinking with image" research, we ultimately adopt only a mild constraint (the segment reward), which encourages the model to invoke segment re-listening without being overly prescriptive about the reasoning trajectory. **Empirically, this formulation has proven both effective and robust**: it successfully increases the model's propensity to reference audio segments (as validated in Table 11 of the updated manuscript), and notably, we do not observe the anticipated reward hacking in practice. This iterative process leads to the final reward composition employed in our approach.
> >
> > Furthermore, **the feasibility of this reward design is strongly corroborated by the comprehensive quantitative results on segment localization and per-segment analysis (as in General Response 2)**. This empirical finding demonstrates that the model, under the current reward scheme, not only revisits segments but does so in a manner that is accurate and beneficial for the reasoning process, thereby validating our design choice.
> >
> >
> > >**W4. In evaluation, the paper also does not evaluate whether the reasoning process correctly extracts useful segments and conducts reasonable thinking for the segment. In my view, there are two ways to fix this problem. First, explicitly parse the segments and per-segment analysis from the outputs and query a good LALM for verification. Second, one can construct a real or synthetic testset with known, accurate time annotations and see if the model can output exactly the same segments and desirable analysis.**
> >
> > Please refer to our response to **W1**.
> >
> > >**Q1. Does the framework apply to complex cases where the model has to listen to several distinct segments at the same time?**
> >
> > Thank you for this insightful question. Our current reasoning framework does not support listening to multiple segments in strict simultaneity. However, **it can achieve a functionally similar outcome through a sequential yet rapid manner**. In our design, each [seg] (<> can not be correctly shown on OpenReview) tag pair only triggers the re-listening of one segment. We observe that the trained model learns to chain these calls when needed, generating a new [seg] tag pair immediately after processing the previous one. Consequently, the subsequent reasoning is conditioned on the accumulated encodings of all recently re-listened segments, allowing the model to perform a de facto integrated analysis across multiple segments even without true parallel re-listening.
> >
> > To further illustrate this capability, **we also include two additional representative cases in Appendix M of the updated manuscript**, where Echo performs multiple, rapid successive re-listening steps to synthesize information and arrive at the correct answer. We kindly invite you to review these examples.

---

> > > ### Author Response · Authors · 2025-11-20
> > > **Response to Reviewer Dkpf (3/3)**
> > >
> > > >**Q2. The segment appending operation seems to impact the model's performance on long audio understanding. How to address this efficiency issue?**
> > >
> > > In our original manuscript, we had not analyzed the model's performance and efficiency by audio length. To address this gap, we have now included a corresponding discussion in Appendix K of the updated manuscript, and the key findings are summarized below.
> > >
> > > | Model | 1-10s (173 Samples) | 11-20s (356 Samples) | 21-30s (468 Samples) | 31-60s (3 Samples) |
> > > |---|:---:|:---:|:---:|:---:|
> > > | Base Model | 52.02 | 52.25 | 50.43 | 33.33 |
> > > | Echo | 67.53 | 70.22 | 67.31 | 100.00 |
> > >
> > > First, regarding performance on long audio, the table above details the model accuracy across different audio lengths on MMAR. Except for three overly lengthy cases, Echo demonstrates consistent proficiency in comprehending audio of varying durations, with a slight performance advantage in the 11-20 second range. **These results confirm that Echo maintains high accuracy on long audio (21-30s) understanding.**
> > >
> > > |  |  | 1-10s |  |  | 11-20s |  |
> > > |---|:---:|:---:|:---:|:---:|:---:|:---:|
> > > |  | Length (word) | Latency (s) | Latency/Length | Length (word) | Latency (s) | Latency/Length |
> > > | Base Model | 65.57 | 1.18 | 0.0180 | 64.32 | 1.09 | 0.0169 |
> > > | Cold-Start Model | 107.12 | 1.82 | 0.0170 | 112.16 | 1.99 | 0.0177 |
> > > | Echo | 96.39 | 1.99 | 0.0206 | 102.90 | 2.09 | 0.0203 |
> > > |  |  | **21-30s** |  |  | **All Samples** |  |
> > > |  | Length (word) | Latency (s) | Latency/Length | Length (word) | Latency (s) | Latency/Length |
> > > | Base Model | 71.31 | 1.26 | 0.0177 | 67.95 | 1.18 | 0.0174 |
> > > | Cold-Start Model | 124.10 | 2.16 | 0.0174 | 117.06 | 2.04 | 0.0174 |
> > > | Echo | 112.99 | 2.20 | 0.0195 | 107.40 | 2.12 | 0.0197 |
> > >
> > > Secondly, regarding the efficiency issue, the table above summarizes the computational efficiency of different models. We found that the primary source of increased latency for longer audio is the longer response length. Since this has been verified in Table 3 of the manuscript, that increased response length emerges as a natural consequence of SFT or RL, we further introduce the latency-to-length ratio to isolate the additional computational overhead specifically attributable to audio-interleaved reasoning. The analysis reveals that the segment-appending operation incurs only about a 13\% increase in this ratio, and this overhead remains stable across different audio durations. **These results indicate that the proposed reasoning strategy maintains acceptable computational efficiency, even when handling long audio.**
> > >
> > > >**Q3. How do you filter inaccurate training samples in Fig 3, especially if the original timestamps are very inaccurate (such as some inaccurately annotated samples in AudioSet-SL)?**
> > >
> > > Please refer to our response to **W2**.
> > >
> > > >**Q4. How does the proposed reward make sure the per-segment reasoning is correct, not hallucination?**
> > >
> > > Please refer to our response to **W3**.
> > >
> > > >**Q5. Can you evaluate the segmenting accuracy and per-segment analysis quality in order to further justify your framework?**
> > >
> > > Please refer to our response to **W1**.

---

> > > > ### Comment · Reviewer_Dkpf · 2025-11-28
> > > > **Response to the authors**
> > > >
> > > > I appreciate the response from authors and the additional experiments that verify the effectiveness of the proposed method. The response addresses most of my concerns.
> > > >
> > > > I agree with the claim that *"many audio QA tasks permit a broad range of plausible reasoning paths"* and I recommend the authors to include a more detailed discussion with respect to this intuition, as this could help readers understand the nature of current AQA benchmarks better. Especially, it would be nice to include some qualitative examples on different reasoning paths and see how these reasoning paths inherit from the knowledge in the original base LLM.
> > > >
> > > > **I think the current manuscript justifies a positive score of 6.**

---

> > > > > ### Author Response · Authors · 2025-11-29
> > > > > **Further Response to Reviewer Dkpf**
> > > > >
> > > > > We want to express our sincere gratitude for your additional effort in reviewing our responses. We greatly appreciate the further suggestions you raised. In response, **we have updated the manuscript by incorporating qualitative examples in Appendix J.2 that illustrate multiple valid reasoning paths leading to the correct answer, along with additional analysis**. Your thoughtful and constructive feedback throughout the discussion has been invaluable, and it has meaningfully strengthened the quality and clarity of our manuscript.

---

### Official Review · Reviewer_uQ9G · 2025-11-03

**Soundness:** 3
**Presentation:** 3
**Contribution:** 4
**Rating:** 8
**Confidence:** 4

**Summary:**

The authors develop an  LALM to perform audio-interleaved reasoning, and report impressive results on existing benchmarks. The model actively revisits audio segments during reasoning, rather than relying on a single  encoding. Training is enhanced by adding an RL stage.  They also create data generation framework using  captions generated via Qwen2.5-Omni and reasoning prompts via DeepSeek-R1.

**Strengths:**

Strong results on the latest benchmarks.
Novel two stage training/inference, bringing RL and Audio revisiting into mainstream of LALMs
Details of dataset creation and frameworks used.

**Weaknesses:**

I would like a stronger commitment to releasing the code than the vague one of releasing it "in the future"

Examples of which segments are revisited would have been good.

Some audio examples would be good.

The biological basis argument could be strengthened.

**Questions:**

you mention the possibilit of bias introduced through the use of LLMs for generating data. Can you amplify this comment?

---

> ### Author Response · Authors · 2025-11-20
> **Response to Reviewer uQ9G**
>
> We sincerely appreciate your positive feedback and valuable comments! Below, we present detailed responses to the weaknesses (W) and questions (Q).
>
> >**W1. I would like a stronger commitment to releasing the code than the vague one of releasing it "in the future".**
>
> **We are fully committed to releasing the model, data, and code once the submission is deanonymized, independent of the final acceptance decision.** The release is currently pending our institution's compliance review, and we will make it public at the earliest possible opportunity thereafter.
>
> >**W2. Examples of which segments are revisited would have been good.**
>
> Thank you for the constructive suggestion. In response, **we have added further case visualizations in Appendix M of the updated manuscript.** These new figures illustrate the segments Echo revisits to handle scenarios that require both rapid, successive segment revisiting and the identification of key information located beyond the initial 10-second audio window.
>
> Furthermore, to complement these qualitative results, we also conduct a comprehensive quantitative analysis of revisited segments, which is presented in **General Response 2** and elaborated in Appendix I of the manuscript. Together, these additions provide a multifaceted evaluation of the revisiting behavior, and we kindly invite you to consult these contents.
>
> >**W3. Some audio examples would be good.**
>
> In parallel with adding the case visualizations for **W2**, we further enhance **Appendix M** by incorporating comprehensive audio descriptions. These descriptions, generated with the assistance of Gemini-2.5-Pro and subsequently refined manually, provide rich context for each case. We agree that this addition significantly improves the clarity and utility of visualizations, and we are grateful for the suggestion.
>
> >**W4. The biological basis argument could be strengthened.**
>
> Upon further literature review, we have cautiously refined the biological argument. The strengthened narrative now reads (references are omitted): *"In contrast, human auditory cognition involves cyclic re-listening of salient acoustic segments, driven by the interaction between auditory working memory and top-down attentional control. This mechanism enables humans to iteratively refine internal auditory representations and achieve more grounded and accurate decision-making."*
>
> We agree that this formulation provides a more compelling motivation for our work, and we have updated the manuscript accordingly (Lines 72-76).
>
> >**Q1. You mention the possibility of bias introduced through the use of LLMs for generating data. Can you amplify this comment?**
>
> Thank you for prompting us to elaborate on this important point. In response, we have expanded our discussion in the Ethics Statement to provide a more detailed and specific explanation of the potential biases.
>
> *Specifically, as revealed by [1], DeepSeek-R1 exhibits systematic "rumination", characterized by repeatedly revisiting similar problem framings rather than exploring diverse reasoning trajectories. The same study also highlights that DeepSeek-R1 may generate harmful or unsafe reasoning traces, which could, in turn, be unintentionally distilled into downstream models during training.*
>
> We have updated the manuscript in Lines 492-496 with this more concrete statement to ensure transparency regarding this limitation.
>
> **References**
>
> [1] DeepSeek-R1 Thoughtology: Let's Think about LLM Reasoning. arXiv 2025.

---

### Official Review · Reviewer_rXCQ · 2025-11-06

**Soundness:** 3
**Presentation:** 3
**Contribution:** 3
**Rating:** 4
**Confidence:** 5

**Summary:**

This paper introduces Echo, a Large Audio Language Model trained for audio-interleaved reasoning: the model decodes <seg>start,end</seg> tags; whenever such a tag is generated, inference is paused and the corresponding raw audio segment is inserted back into the context, so the model “thinks with audio,” not just about it. Training is two-stage: (1) SFT to make the model reference informative segments in CoT; (2) RL with verifiable rewards to encourage correct formatting, consistency after </seg>, accuracy, and segment usage. A synthetic, LLM-curated pipeline builds the training sets (EAQA-SFT 75.9k with CoT; EAQA-RL 21.9k without), using Qwen2.5-Omni for audio descriptions/transcripts/music analysis and DeepSeek-R1 to synthesize and filter QA-CoT. Evaluations on MMAR, MMAU-mini, and MMAU show Echo beating open and proprietary baselines on average accuracy, with analyses on attention allocation and segment coverage.

**Strengths:**

- Turning audio from a static pre-embedding into an active element during reasoning is simple, intuitive, and well thought out with the <seg> scheme and inference adaptation.
- The SFT to RL pipeline is coherent; rewards are explicitly specified (format, "consistency" after </seg>, accuracy, + a segment-use bonus), making the approach reproducible in principle.
- The paper details how temporal metadata + Qwen2.5-Omni descriptors feed DeepSeek-R1 for QA-CoT synthesis and filtering into SFT and RL splits.
- Results on MMAR and MMAU are good; Echo tops macro-average accuracy among peers and even improves o ver some proprietary models on MMAR.
- Good ablation study with the recipe comparison table (A --> B --> C --> D) attributes gains, shows initial interleaving hurt without RL, then recovers to best accuracy-which is an useful insight.

**Weaknesses:**

- No human evaluation of the thinking chain.
- No qualitative analysis of the data being generated by the LLMs for training, this might induce bias or the training data might also contain some hallucinated data.
- EAQA-SFT CoTs are constrained by source temporal metadata (the paper later notes SFT annotations limited to first 10s), yet claimed generalization beyond 10s is argued only by aggregate coverage stats; no targeted stress-tests show failure modes when informative cues cluster late. The generalization claim needs more and stronger evidence.
- Since the model outputs thinking chains, the authors should specify whether they use the thinking chain as well during evaluating against MMAU and MMAR. The thinking traces might contain multiple answers and regex/string based answer matching might get compromised in this setting.
- Although the RL pipeline and rewards are explicitly specified, the "consistency" reward checks whether the token after </seg> is capitalized or <; this is a formatting heuristic, not a perceptual constraint. The segment reward grants +0.5 if at least one segment is referenced and the answer is correct, this is easy to satisfy by superficial/random segment references once the model learns the distribution of answer keys. There’s no penalty for irrelevant segments or when a model generates lots of <seg>start,end</seg> tags across the timeline beyond minor format checks.

**Questions:**

See weakness section.

---

> ### Author Response · Authors · 2025-11-20
> **Response to Reviewer rXCQ (1/3)**
>
> We sincerely appreciate your thoughtful effort in conducting a valuable review! Below, we present detailed responses to the weaknesses (**W**).
>
> >**W1. No human evaluation of the thinking chain.**
>
> We kindly refer you to our **General Response 2**. In this response window, we have supplemented the evaluation of the reasoning chain by not only comparing the model's segment localization performance against **human annotations**, but also adopting an LLM-as-a-judge approach to assess the relevance of its per-segment analysis to the audio content, as well as its contribution to the final conclusion. The results demonstrate that **Echo exhibits high proficiency in accurately locating audio segments and performs the corresponding analysis as intended**. These results can also be found in Appendix I of the updated manuscript.
>
> >**W2. No qualitative analysis of the data being generated by the LLMs for training, this might induce bias or the training data might also contain some hallucinated data.**
>
> We acknowledge your valid concern regarding potential noise and hallucinations in LLM-synthesized data. Our data annotation pipeline reduces such noise through two complementary mechanisms.
>
> First, during the synthesis of QA-CoT triplets, the LLM (DeepSeek-R1) is provided with audio information from multiple sources, including temporal metadata and predictions from Qwen2.5-Omni. **These sources implicitly cross-validate one another, allowing the LLM to assess the relative reliability of different cues**. This mechanism promotes a more selective use of temporal annotations, thereby mitigating the propagation of noise into synthesized data.
>
> Second, **the LLM performs an additional round of re-evaluation to further improve the quality of the synthesized training data**. In this stage, temporal metadata, audio information extracted from Qwen2.5-Omni, and the previously synthesized QA-CoT triplets are jointly provided to the LLM, allowing it to reassess each sample with full contextual evidence. Through explicit prompting, the LLM is instructed to verify answer plausibility, identify ambiguous choices, and detect unsupported or hallucinated CoT steps. As a result, the annotation noise and hallucinated reasoning are further suppressed in the final training sets. **Two representative examples are illustrated in Appendix F.2 of the updated manuscript.**
>
> **Furthermore, these two mechanisms can also be validated through quantitative evidence.** The table below specifically substantiates the effect of the first mechanism (implicit cross-validation) by comparing the impact of different audio extractors on data quality and downstream performance:
>
> | Audio Extractor | MMAR (Acc%) | MMAU-mini (Acc%) |
> |---|:---:|:---:|
> | Qwen2.5-Omni | 69.99 | 80.41 |
> | Qwen2-Audio | 63.91 | 75.41 |
>
> The results show that using a more capable audio extractor (to extract audio description, speech transcription, and musical analysis) leads to significantly higher model accuracy on both benchmarks. **This demonstrates how enhanced audio information enables more effective implicit cross-validation during triplet synthesis, thereby reducing noise propagated from temporal metadata and improving the overall quality of the generated data.**
>
> The impact of the second mechanism (explicit re-evaluation) is quantitatively validated in the table below:
>
> | SFT Data | RL Data | MMAR (Acc%) | MMAU-mini (Acc%) |
> |---|---|:---:|:---:|
> | 76k Samples Re-evaluated as High Quality | 22k Samples Re-evaluated as High Quality | 69.99 | 80.41 |
> | 111k Unfiltered Samples (Expect for RL Data) | 22k Samples Re-evaluated as High Quality | 64.67 | 76.19 |
> | 76k Samples Re-evaluated as High Quality | 22k Samples Re-evaluated as Low Quality | 67.03 | 77.40 |
>
> From the results, the model achieves optimal results when both SFT and RL data consist solely of high-quality samples (first row). Introducing low-quality samples into SFT data causes a clear performance drop (second row), confirming that re-evaluation effectively reduces CoT noise and hallucinations. When RL data is replaced with low-quality samples (a cost-efficient alternative to scaling RL), performance also declines (third row). **This highlights the effectiveness of re-evaluation in mitigating data noise.** The complete versions of these two experiments are provided in Appendix F.1&F.2 in the updated manuscript.

---

> > ### Author Response · Authors · 2025-11-20
> > **Response to Reviewer rXCQ (2/3)**
> >
> > >**W3. EAQA-SFT CoTs are constrained by source temporal metadata (the paper later notes SFT annotations limited to first 10s), yet claimed generalization beyond 10s is argued only by aggregate coverage stats; no targeted stress-tests show failure modes when informative cues cluster late. The generalization claim needs more and stronger evidence.**
> >
> > We agree that our claim regarding model generalization would benefit from stronger supporting evidence. However, due to the high cost and inherent subjectivity of manual annotation, it is currently infeasible for us to collect a sufficiently large test set in which informative cues are consistently clustered in later segments. As an alternative, we present two additional quantitative analyses to validate the model's generalizability on longer audio.
> >
> > | Model | 1-10s (173 Samples) | 11-20s (356 Samples) | 21-30s (468 Samples) | 31-60s (3 Samples) |
> > |---|:---:|:---:|:---:|:---:|
> > | Base Model | 52.02 | 52.25 | 50.43 | 33.33 |
> > | Echo | 67.53 | 70.22 | 67.31 | 100.00 |
> >
> > The table above details the model accuracy across different audio lengths on MMAR. Except for three overly lengthy cases, Echo demonstrates consistent proficiency in comprehending audio of varying durations, with a slight performance advantage in the 11-20 second range. **These results provide straightforward support for its ability to maintain accuracy when processing longer audio.**
> >
> > | Audio Duration | Intersect with [0-10s] | Intersect with [10.1-20s] | Intersect with [20.1-30s] |
> > |---|:---:|:---:|:---:|
> > | 0-10s (173 Samples) | 99.42 | - | - |
> > | 11-20s (356 Samples) | 97.75 | 24.16 | - |
> > | 21-30s (468 Samples) | 90.38 | 37.18 | 9.40 |
> > | 31-60s (3 Samples) | 100.0 | 66.67 | 0 |
> >
> > The table above reports the proportion of re-listened segments that overlap with different time intervals, illustrating the temporal coverage of the Echo's re-listening behavior across audio of varying lengths. For example, in 11–20 second audios, 97.75% responses re-listen to segments overlapping with the [0–10s] range, and 24.16% also involve the [10.1–20s] interval. These results provide an intuitive view of the distribution of re-listened audio segments. Although segments beyond the 10-second mark are accessed less frequently than those within the first 10 seconds, the model still retrieves excerpts longer than 10 seconds, and even beyond 20 seconds, in a non-trivial fraction of cases. **This fine-grained analysis strengthens the evidence for Echo's generalization capability in segment retrieval and helps explain its maintained accuracy on longer audio, such as those lasting 21–30 seconds.** These experiments are also included in Appendix K of the updated manuscript.
> >
> > Complementing this quantitative perspective, **Appendix M in the updated manuscript provides qualitative case studies where the critical clues are concentrated in the latter parts of the audio**. The results clearly show that Echo demonstrates the ability to reference segments beyond 10 seconds and leverages them to form rational analyses leading to correct conclusions. This outcome further verifies Echo's generalization in accessing and utilizing audio information across extended temporal contexts.
> >
> > >**W4. Since the model outputs thinking chains, the authors should specify whether they use the thinking chain as well during evaluating against MMAU and MMAR. The thinking traces might contain multiple answers and regex/string-based answer matching might get compromised in this setting.**
> >
> > In evaluation, we consider an answer correct if and only if the content enclosed within the [answer] (<> can not be correctly shown on OpenReview) tags matches exactly with the ground-truth, ignoring case and special characters. This approach guarantees that each model response is mapped to at most one answer, preventing any unfair comparisons. We have clarified this in Lines 317-320 of the updated manuscript to eliminate potential ambiguity.

---

> > > ### Author Response · Authors · 2025-11-20
> > > **Response to Reviewer rXCQ (3/3)**
> > >
> > > >**W5 (1). Although the RL pipeline and rewards are explicitly specified, the "consistency" reward checks whether the token after [/seg] is capitalized or <; this is a formatting heuristic, not a perceptual constraint.**
> > >
> > > We design the consistency reward primarily to encourage responses that adhere to fundamental syntax patterns in its analysis. As demonstrated in Figure 12 of the manuscript, models trained without this reward frequently exhibit unstable behavior after re-listening to audio segments, often skipping analytical steps and jumping directly to the final answer. The introduction of this reward effectively mitigates this issue.
> > >
> > > Upon reflection, we acknowledge that the term "consistency" could indeed cause confusion. To eliminate any potential ambiguity, we consider renaming it to "syntax reward" in the future manuscript.
> > >
> > > >**W5 (2). The segment reward grants +0.5 if at least one segment is referenced and the answer is correct, this is easy to satisfy by superficial/random segment references once the model learns the distribution of answer keys. There's no penalty for irrelevant segments or when a model generates lots of [seg]start, end[/seg] tags across the timeline beyond minor format checks.**
> > >
> > > Our choice of reward implementation is driven primarily by empirical observations. During exploration, we experiment with more explicit forms of process supervision, such as rewards based on textual semantic matching or segment IoU. However, **we find that these do not yield consistent performance gains and occasionally destabilize the RL training process.** We hypothesize that this occurs because many audio QA tasks permit a broad range of plausible reasoning paths, and imposing explicit constraints on these paths may unduly restrict the model's exploration.
> > >
> > > Drawing on insights from "thinking with image" research, we ultimately adopt only a mild constraint (the segment reward), which encourages the model to invoke segment re-listening without being overly prescriptive about the reasoning trajectory. **Empirically, this formulation has proven both effective and robust**: it successfully increases the model's propensity to reference audio segments (as validated in Table 11 of the updated manuscript), and notably, we do not observe the anticipated reward hacking (e.g., superficial or random segment references mentioned in the weakness) in practice. This iterative process leads to the final reward composition employed in our approach.
> > >
> > > Furthermore, **the feasibility of this reward design is strongly corroborated by the comprehensive quantitative results on segment localization and per-segment analysis (as in General Response 2)**. This empirical finding demonstrates that the model, under the current reward scheme, not only revisits segments but does so in a manner that is accurate and beneficial for the reasoning process, thereby validating our design choice.

---

### Author Response · Authors · 2025-11-20
**General Response 1**

We sincerely thank all the reviewers for their insightful critiques and valuable suggestions regarding our manuscript. Your insights have been invaluable in enhancing the overall quality of our work. We have responded to the reviews individually and **updated a new manuscript version**.

We also wish to express our particular appreciation for the reviewers' recognition of our strengths, including the novelty of the audio-interleaved reasoning paradigm, the benefits of the data generation pipeline, the comprehensive empirical results, and the potential for inspiring future works. This positive feedback is greatly encouraging.

In the following, we summarize the main improvements made to the revision, and we remain fully available for any additional inquiries.

Main manuscript (all changes are marked in blue):
- Lines 72-76: Strengthened biological basis argument (@uQ9G).
- Lines 77-78: Rephrasing of the advantage of the proposed audio-interleaved reasoning (@H3Dr).
- Line 284 (Table 1): Performance of Gemini-2.5-Pro on MMAR (@jS6b).
- Lines 317-320: Clarification for evaluation details (@rXCQ, H3Dr).
- Lines 492-496: Details of potential biases introduced through LLM-generated data (@uQ9G).

Appendix (titles of additional sections are marked in blue in the appendix menu):
- Appendix F.2: Mitigation of noises in the source metadata (@rXCQ, uQ9G, Dkpf, H3Dr).
- Appendix F.3: Influence of synthesizer, re-evaluator, and audio extractor (@jS6b).
- Appendix F.4: Influence of the re-evaluation process (@rXCQ, Dkpf, H3Dr).
- Appendix I: Quantitative analysis of re-listened segments (@rXCQ, uQ9G, Dkpf, H3Dr, jS6b).
- Appendix J.2: The decision against direct process supervision (@rXCQ, Dkpf, H3Dr).
- Appendix K: Generalizability to long audio (@rXCQ, Dkpf).
- Appendix M: More case visualization (@uQ9G, Dkpf).

---

> ### Author Response · Authors · 2025-11-20
> **General Response 2 (1/2)**
>
> In preparing our point-by-point responses, we note a converging emphasis among several reviewers on the need to evaluate the reasoning chain, particularly regarding the accuracy of segment localization and the utility of per-segment analysis. We agree that such evaluations would strengthen the manuscript, and we are deeply grateful for this guidance. Inspired by Reviewer Dkpf's constructive suggestions, we present the additional quantitative results in this response window to address these shared concerns.
>
> ## Evaluation for Segment Localization
>
> To quantitatively evaluate the accuracy of Echo in locating salient segments, we first introduce two sample-level evaluation metrics: *segment precision* and *segment coverage*. Both metrics rely on the Intersection over Union (IoU) between the predicted (re-listened) segments and the annotated segments. For two segments $A_{s_1:e_1}$ and $A_{s_2:e_2}$, where $s_1$, $s_2$, denote start times and $e_1$, $e_2$ denote end times, the IoU is calculated as:
> \begin{equation}
> \text{IoU}(A_{s_1:e_1}, A_{s_2:e_2})=\frac{|(s_1, e_1)\bigcap(s_2, e_2)|}{|(s_1, e_1)\bigcup(s_2, e_2)|}.
> \end{equation}
> Based on this, *segment precision* is defined as the proportion of predicted segments in a sample that have an $\text{IoU}\ge\rho$ with at least one annotated segment, reflecting localization accuracy. Let the predicted segments for a sample be $\mathcal{P}=\lbrace p_1,p_2,\dots,p_{n} \rbrace$, and the set of annotated segments be $\mathcal{A}=\lbrace a_1,a_2,\dots,a_{m}\rbrace$. *Segment precision* at threshold $\rho$ is defined as:
> \begin{equation}
> P_s(\rho)=\frac{1}{|\mathcal{P}|}
> {\bigl|\lbrace p_i \in \mathcal{P} \mid \exists\, a_j \in \mathcal{A},\ \text{IoU}(p_i,a_j)\ge \rho \rbrace\bigr|}.
> \end{equation}
> Conversely, *segment coverage* is defined as the proportion of annotated segments that have an $\text{IoU}\ge\rho$ with at least one predicted segment, indicating how well the predicted segments cover the annotated segments. *Segment coverage* at threshold $\rho$ is defined as:
> \begin{equation}
> C_s(\rho)=\frac{1}{|\mathcal{A}|}
> {\bigl|\lbrace a_i \in \mathcal{A} \mid \exists\, p_j \in \mathcal{P},\ \text{IoU}(a_i,p_j)\ge \rho \rbrace\bigr|}.
> \end{equation}
> We visually illustrate these metrics with Figure 11 in the updated manuscript. Based on these metrics, we conduct a comprehensive evaluation of the model's proficiency in localizing audio segments using both our manually annotated and existing temporal metadata. In the absence of comparable baselines, we compute the average *segment precision* and *segment coverage* for both the cold-start model and Echo across three subsets: randomly selected samples, correctly answered samples, and incorrectly answered samples. This analysis offers insight into how RL influences segment-localization ability and how the quality of this localization, in turn, impacts the correctness of the final responses.
>
> | Model | Split | Ps(0.3) | Cs(0.3) | Ps(0.5) | Cs(0.5) |
> |---|---|---|---|---|---|
> | Cold-Start Model | Random | 88.32 | 70.62 | 81.21 | 59.68 |
> | Cold-Start Model | Correctly Answered | 91.03 | 78.29 | 83.85 | 67.75 |
> | Cold-Start Model | Incorrectly Answered | 86.79 | 63.85 | 78.19 | 52.32 |
> | Echo | Random | 92.47 | 73.22 | 84.41 | 62.04 |
> | Echo | Correctly Answered | 94.25 | 79.26 | 85.38 | 68.98 |
> | Echo | Incorrectly Answered | 91.84 | 67.99 | 82.35 | 56.93 |
>
> On MMAR, we select 50 correctly answered samples, 50 incorrectly answered samples, and 50 non-overlapping random samples, and manually annotate the audio segments that are critical for answering each question. These human annotations serve as the reference for evaluating average *segment precision* and *segment coverage*, with the results summarized in the table above.
>
> **Echo achieves high performance on both metrics, indicating strong proficiency in segment localization.** For correctly answered samples, Echo consistently exceeds its performance on incorrect samples in both precision and coverage, revealing a clear positive correlation between accurate segment localization and effective audio comprehension. The gap in *segment coverage* is particularly notable, suggesting that some model errors arise not from mislocalizing segments but from failing to retrieve essential regions of the audio.
>
> The cold-start model exhibits localization ability that is already comparable to Echo's, though RL still provides measurable gains. This pattern suggests that **outcome-based rewards implicitly improve segment-grounding behavior, even without explicit supervision on segment localization.**

---

> > ### Author Response · Authors · 2025-11-20
> > **General Response 2 (2/2)**
> >
> > Since our manual annotations inevitably introduce some degree of subjectivity, we further validate our conclusions using existing temporal metadata. The AudioGrounding dataset [1], originally designed for the text-to-audio grounding task, contains manually annotated sound event timestamps with relatively low noise. However, it does not provide QA pairs by default, so we apply our proposed data generation pipeline to construct 500 QA samples from its audio and event annotations.
> >
> > | Model | Split | Ps(0.3) | Cs(0.3) | Ps(0.5) | Cs(0.5) |
> > |---|---|---|---|---|---|
> > | Cold-Start Model | All 500 Samples | 84.37 | 60.27 | 77.67 | 51.89 |
> > | Cold-Start Model | Correctly Answered | 90.53 | 66.70 | 81.98 | 55.01 |
> > | Cold-Start Model | Incorrectly Answered | 75.02 | 53.16 | 75.04 | 45.77 |
> > | Echo | All 500 Samples | 89.34 | 64.83 | 81.23 | 55.58 |
> > | Echo | Correctly Answered | 91.65 | 69.71 | 83.88 | 59.75 |
> > | Echo | Incorrectly Answered | 86.19 | 60.04 | 80.19 | 51.86 |
> >
> > In addition to overall evaluating all constructed samples, we also sample 50 correctly answered and 50 incorrectly answered cases, with results reported in the table above. The overall trend is consistent with our earlier findings, though all metrics are slightly lower, especially *segment coverage*. This decline occurs because the segment labels in AudioGrounding are not filtered for task relevance, thus including segments unrelated to the QA task, which reduces measured coverage. Despite such differences, **the consistent pattern across datasets validates our analysis.**
> >
> > ## Evaluation for Per-Segment Analysis
> >
> > To assess the plausibility of the model's analysis for each referenced segment, we adopt an LLM-as-a-judge evaluation. For each segment reference, we first extract the full sentence in which it appears. This sentence, together with the corresponding Audio-QA pair and the model's complete response, is then provided to an advanced LALM (Gemini-2.5-Pro in our implementation). The model is instructed to evaluate whether the analysis expressed in the sentence is relevant to the content of the referenced audio segment (represented by the *relevance* metric) and whether it contributes meaningfully to the final answer (represented by the *contribution* metric). The prompt is reported in the revised manuscript.
> >
> > | Model | Split | Relevance | Contribution |
> > |---|---|---|---|
> > | Cold-Start Model | Random | 86.0 | 72.0 |
> > | Cold-Start Model | Correctly Answered | 86.0 | 78.0 |
> > | Cold-Start Model | Incorrectly Answered | 82.0 | 64.0 |
> > | Echo | Random | 94.0 | 84.0 |
> > | Echo | Correctly Answered | 94.0 | 88.0 |
> > | Echo | Incorrectly Answered | 92.0 | 72.0 |
> >
> > We treat both *relevance* and *contribution* as segment-level metrics, assigning each a binary judgment (1 or 0). These metrics are computed on the segments drawn from the previously selected 150 MMAR samples, and the aggregated results are reported in the table above.
> >
> > The results indicate that, regardless of whether the final answer is correct or incorrect, **Echo consistently produces highly relevant segment-level analyses, substantially outperforming the cold-start model.** This highlights the advantage of audio-interleaved reasoning over an initial audio-grounded approach: re-listening to key segments enables Echo to deliver more accurate and context-aware interpretations. In contrast, the *contribution* metric exhibits a clear separation between correctly and incorrectly answered samples. Together with the findings from localization evaluation, this pattern suggests a potential failure mode: Echo may sometimes anchor its reasoning on suboptimal segments. Even when the analysis of such segments is itself accurate, their weak alignment with the question can still mislead the final decision.
> >
> > Overall, **Echo attains high performance on both *relevance* and *contribution*, confirming that the model is capable of producing precise and reliable per-segment analyses as intended.**
> >
> > **References**
> >
> > [1] Text-to-audio grounding: Building correspondence between captions and sound events. ICASSP 2021.

---

### Author Response · Authors · 2025-11-27
**Inquiry on Discussion Period Feedback**

Dear reviewers,

We would like to once again express our sincere appreciation for the time and effort you have devoted to reviewing our manuscript. As the discussion period is approaching its end in a week, we kindly wish to check whether our responses have addressed your concerns. If any questions or concerns remain, we would greatly appreciate hearing from you at your earliest convenience, so that we can address them promptly within the discussion window.

---

### Author Response · Authors · 2025-11-30
**Summary of the Rebuttal Period**

We would like to express our sincere gratitude to the area chair and all reviewers for the time and effort you have devoted during the rebuttal period. In light of the unexpected incident that interrupted the discussion, we would like to provide a brief summary of the key points regarding our manuscript.

In this work, we propose an audio-interleaved reasoning paradigm, along with the corresponding training framework and data generation pipeline, to advance LALMs in audio comprehension. The paradigm treats audio as active reasoning components rather than static context, and we demonstrate its effectiveness through extensive experiments on MMAU and MMAR.

In the initial reviews, **all reviewers recognized the novelty of this reasoning paradigm; the value of our training and data generation pipelines; and the strong empirical gains and comprehensive analytical experiments**. Several reviewers also highlighted the potential contributions to the community and the clarity of the presentation. This is reflected in the Soundness, Presentation, and Contribution scores, where all of our ratings are at least 3.

Regarding the concerns raised, three questions are mostly shared among the reviewers, which are primarily for further validation and clarification:
- Although the overall performance is strong, the manuscript lacks quantitative evaluations of the re-listened segments during the reasoning process.
- The data generation pipeline is not sufficiently detailed in explaining how noise is mitigated.
- The rationale for not directly supervising segment re-listening in the reward design is unclear.

Additional concerns include the generalizability to long audio, further analysis of efficiency, and requests for clearer explanations and quantitative examples. These comments are extremely valuable. In response, **we have conducted targeted experiments for each of them and incorporated the corresponding analyses into the updated manuscript (detailed in General Response 1).** This has significantly strengthened both the completeness and coverage of the manuscript.

Up to the point when the discussion was interrupted, one reviewer (Dkpf) had responded to our rebuttal and indicated that **most of their concerns, including the three shared issues, had been addressed**, and that they intended to raise their score. This makes us cautiously optimistic that our responses may have also resolved similar concerns held by the other reviewers.

We sincerely hope that the clarifications and revisions we have provided adequately address the reviewers’ concerns. We are grateful for the thoughtful feedback and constructive suggestions throughout this process. Thank you once again for your careful evaluation of our manuscript.

---

### Meta-Review · Area_Chair_naaZ · 2026-01-06

**Summary:**

Reviewers mainly have concerns on points related to LLM-RL
- Are segment correctly selected?
- Is reasoning needed?
- Quality of synthesized data
- Bad segment is not penalized

In my personal opinion, I like this paper, and understand the reason using same audio encoder in "echo" but believe using a different encoder, or at least with a different flag of encoder will be a game changer, which can be future work.

**Reviewer Concerns:**

- Are segment correctly selected?

Addressed
- Is reasoning needed?

Addressed
- Quality of synthesized data

Partially
- Bad segment is not penalized

This needs more discussion and experiment

**Reviewer Scores:**

Dkpf

Will increase score to accept.

rXCQ

Will increase score to boardline accept.

uQ9G / jS6b

Won't decrease scores

H3Dr

May not change

---

### Decision · Program_Chairs · 2026-01-26

Accept (Poster)